# Teacher-student collaborated multiple instance learning for pan-cancer *PDL1* expression prediction from histopathology slides

Darui Jin [1,2,3], Shangying Liang[1], Artem Shmatko [2], Alexander Arnold[4], David Horst[4,5], Thomas G. P. Grünewald [6,7,8,9] ✉, Moritz Gerstung [2] ✉ & Xiangzhi Bai [1,10,11] ✉

Programmed cell death ligand 1 (PDL1), as an important biomarker, is quantified by immunohistochemistry (IHC) with few established histopathological patterns. Deep learning aids in histopathological assessment, yet heterogeneity and lacking spatially resolved annotations challenge precise analysis. Here, we present a weakly supervised learning approach using bulk RNA sequencing for *PDL1* expression prediction from hematoxylin and eosin (H&E) slides. Our method extends the multiple instance learning paradigm with the teacher-student framework, which assigns dynamic pseudo-labels for intra-slide heterogeneity and retrieves unlabeled instances using temporal ensemble model distillation. The approach, evaluated on 12,299 slides across 20 solid tumor types, achieves a weighted average area under the curve of 0.83 on fresh-frozen and 0.74 on formalin-fixed specimens for 9 tumors with PDL1 as an established biomarker. Our method predicts *PDL1* expression patterns, validated by IHC on 20 slides, offering insights into histologies relevant to PDL1. This demonstrates the potential of deep learning in identifying diverse histological patterns for molecular changes from H&E images.

Inhibitors for the PD1-PDL1 checkpoint have revolutionized cancer therapy in the past decade. In addition to seven anti-PD1/PDL1 monoclonal antibodies (mAbs) currently approved by US Food and Drug Administration (FDA), there are still approximately six thousand mAbs undergoing clinical trials[1–3]. Blockade of PD1-PDL1 interaction notably contributes to activating antitumor immunity and is proven to benefit the treatment of various types of tumors[4–6]. The expression of PDL1 (gene symbol CD274) serves as a biomarker associated with patients' response to anti-PD1/PDL1 mAbs such as pembrolizumab and nivolumab, which acts as the most widely adopted standard for identifying

[1]Image Processing Center, Beihang University, Beijing 102206, China. [2]Division of AI in Oncology, German Cancer Research Center (DKFZ), Heidelberg, Germany. [3]Shen Yuan Honors College, Beihang University, Beijing 100191, China. [4]Charité - Universitätsmedizin Berlin, Institute of Pathology, 10117 Berlin, Germany. [5]German Cancer Consortium (DKTK), partner site Berlin, a partnership between DKFZ and Charité-Universitätsmedizin Berlin, Berlin, Germany. [6]Institute of Pathology, Heidelberg University Hospital, Heidelberg, Germany. [7]Division of Translational Pediatric Sarcoma Research, German Cancer Research Center (DKFZ), German Cancer Consortium (DKTK), Heidelberg, Germany. [8]Hopp Children's Cancer Center (KiTZ) Heidelberg, Heidelberg, Germany. [9]National Center for Tumor Diseases (NCT), NCT Heidelberg, a partnership between DKFZ and Heidelberg University Hospital, Heidelberg, Germany. [10]State Key Laboratory of Virtual Reality Technology and Systems, Beihang University, Beijing 100191, China. [11]Advanced Innovation Center for Biomedical Engineering, Beihang University, Beijing 100083, China. ✉e-mail: t.gruenewald@kitz-heidelberg.de; moritz.gerstung@dkfz.de; jackybxz@buaa.edu.cn

patient cohorts that are appropriate candidates for PD1-PDL1 immunotherapy[7]. From 2011 to 2021, FDA approvals of 15 immune checkpoint inhibitors were linked with companion PDL1 testing including non-small cell lung cancer (NSCLC) ($N=7$), bladder cancer ($N=3$), triple-negative breast cancer ($N=2$), cervical cancer ($N=2$) and gastric cancer ($N=1$). Further, the outcomes of patients with renal cell cancer, colon cancer and melanoma are also reported linked with PDL1 expression by many reports[8–12].

Currently, PDL1 expression is predominantly quantified by immunohistochemistry (IHC) assays, and some recent research has also indicated a significant correlation between mRNA expression levels and the response of associated monotherapies[13]. Whilst IHC qualification has been successfully used in clinical practice for decades and is considered to be a gold standard for this task, recent analyses show that the interpretation of staining and decision threshold differs for different commercially available platforms and even within the same platform[14]. Along with the subjectivity of pathologists, these factors introduce undesired

inter- and intraobserver variance to the evaluation of staining, limiting the reproducibility. Besides, quantification of mRNA expression using techniques like real-time reverse transcription polymerase chain reaction and IHC tests can be costly and time-consuming[15]. H&E stained slides are one of the most widely used and effective carriers of pathological information, offering a cost-effective and expedient alternative, and are employed in routine pathological assessment of clinical specimens. Developing an efficient and reliable method for estimating PDL1 expression on H&E stained slides, which does not require additional sample preparations, may yield a faster and cheaper diagnostic readout. Further, this process is capable of unraveling histopathological characteristics of PDL1 expression on slides, thereby assisting pathologists in comprehending the gene's expression mechanism and facilitating a more precise interpretation of immune evasion patterns.

The expanding field of computational histopathology may not only automate existing workflows, but also proposes to explore the molecular information based on morphological features via deep

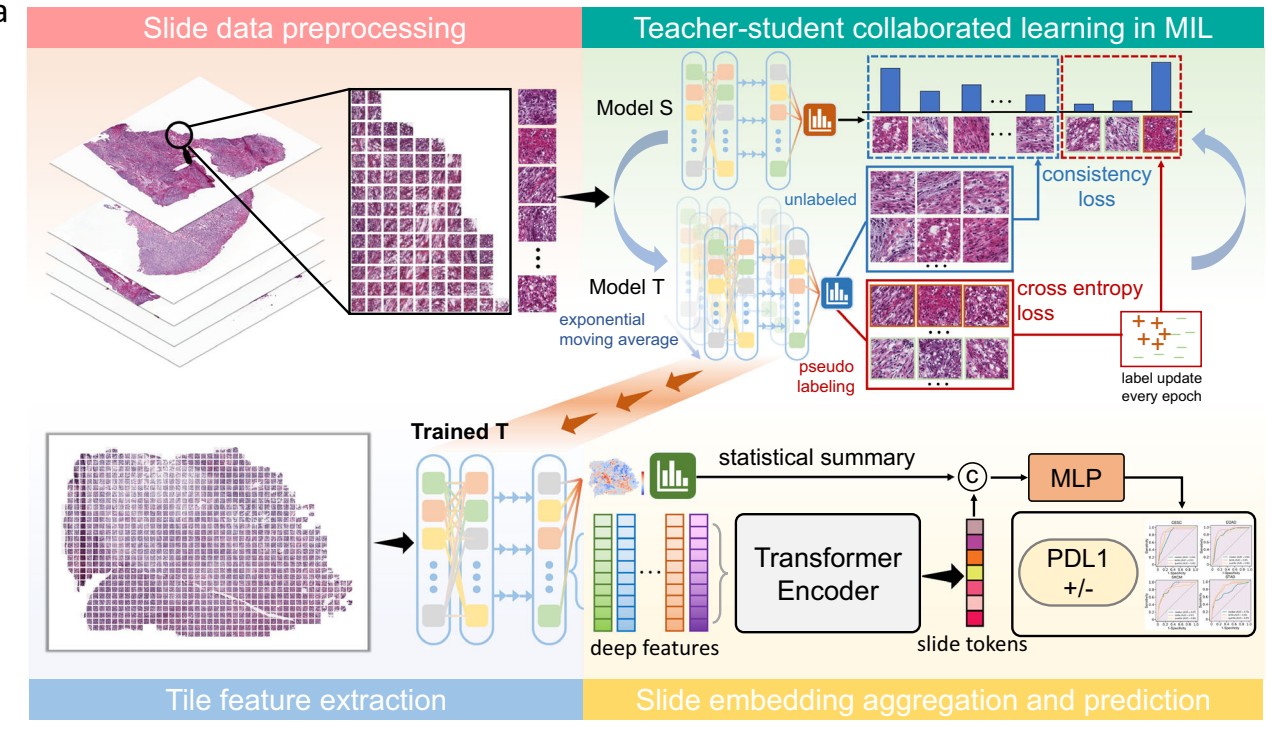

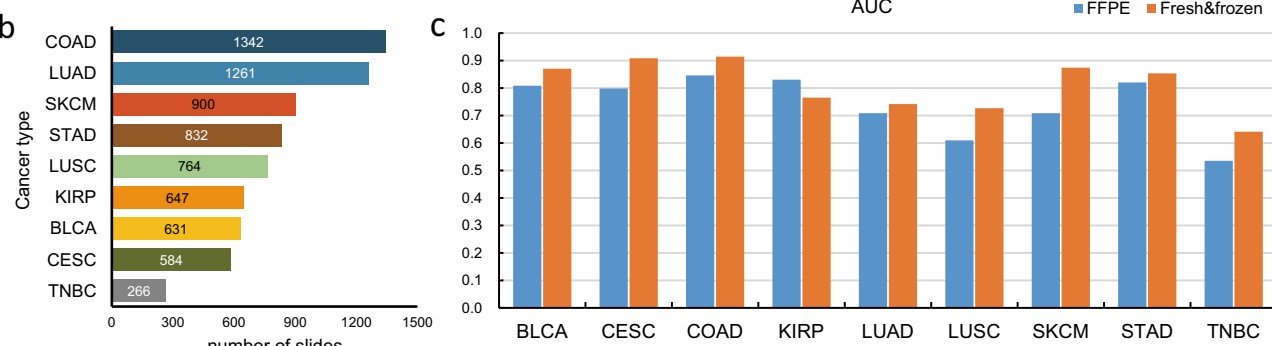

**Fig. 1 | The framework of MILTS and its performance on clinically PDL1-relevant tumors. a** The training and inference workflow of MILTS includes three steps. First, the data of patient cohorts are divided into training set, validation set and test set, followed by patching and random augmentations. Then, obtained tiles are utilized to train the patch-level teacher-student collaborated network in a MIL manner. At last, the trained patch-level teacher model (or student model) works as the extractor of both statistical features and deep features. The deep features of patches in the same slide are further fused into a slide token and combined with the statistical summary of patch-level features to train an MLP classifier which gives the patient-level diagnosis. MIL multiple instance learning, S student, T teacher, C concatenation, MLP multi-layer perceptron. **b** Quantities of slide images of different tumors. **c** Plot illustrating the model's performance on FFPE slides and fresh-frozen slides for the aforementioned tumors, separately. Source data are provided as a Source Data file.

learning[16]. It has exhibited great potential in assisting pathologists on many routine tasks[17] including applications such as mitosis detection[18–21], tissues segmentation[22–25], tumor subtyping and grading[26–29], and biomarker assessment[30–33]. Some studies also reveal that morphotypes are closely associated with genetic alterations in tumors and thus indicative of clinical features and prognosis, which has a chance to redefine the clinical workflows[34–37]. However, the analysis of pathological images is often challenged by the limited availability of reference data, with clinical reports and patient-level diagnoses being the main sources of information[16]. And it is also difficult to accurately capture and spatially resolve transcriptome profiles at the pixel or cell level in whole slide images[38]. The presence of such heterogeneity poses challenges in training deep learning based algorithms, which usually requires accurate references in the form of labeled data[39]. Some studies used manual data annotation and adopted fully supervised learning strategies. For example, in the works of Sha et al.[40] and Shamai et al.[33], PDL1 expression quantification was performed at the tile level by pathologists using paired IHC slides providing an accurate reference for training. However, the dataset size could be constrained due to the labor-intensive nature of annotating such data. Some other approaches have opted to overlook the intra-tumor heterogeneity by assigning slide-level annotations to all content within a slide[31,34,35,41,42]. This strategy works well if the slides exhibit good homogeneity concerning specific properties of interest. However, it can also lead to overfitting and undesired generalization with heterogeneous composition, where inaccurate instance-level labels will either hinder the convergence of the model or result in erroneous recognition of relevant patterns[43]. More recent approaches tend to directly utilize ImageNet[44] pretrained features and incorporate specially designed attention or embedding-based MIL strategies[19,24,29,36,45–48], which significantly accelerates the training but is also more data-hungry. In addition, dimensionality reduction with pretrained features inevitably leads to information loss, since such ImageNet pretrained models are designed to capture general visual patterns without any specific bias towards histopathology-related features or priors. Consequently, in situations where certain details in histopathology images are completely lost or unavailable, reweighting or attention mechanisms have limited effectiveness.

In this work we propose a weakly supervised learning based methodology named MILTS (Teacher-Student collaborated Multiple Instance Learning framework) to leverage the massive amount of tile information and slide-level annotation provided by whole slide H&E images from The Cancer Genome Atlas (TCGA) and The National Cancer Institute's Clinical Proteomic Tumor Analysis Consortium (CPTAC) involving 12,299 slides from 6715 patients across 20 kinds of tumors. MILTS utilizes an iterative, self-refining process to assign labels automatically supporting tile feature extractor training, and combines the statistical summary of tile-level predictions with tokens fused by a transformer to obtain the slide-level embedding for the final prediction. Results demonstrate there exists a salient morpho-transcriptomic link across cancer types, whose treatment landscape and prognosis are associated with PDL1 gene expression, with a weighted average AUC of 0.83 on fresh-frozen and 0.74 on FFPE specimens. Heterogeneous tile-level predictions further provide insights into morphotypes associated with PDL1 hot regions in colon cancer, which include mixed inflammatory stroma with relatively high abundance of eosinophils and a cribriform growth pattern of tumor cells with hyperchromatic nuclei. Model predictions also consistently exhibited a strong positive correlation with corresponding IHC quantification, providing further validation for the findings based on H&E staining. Varying degrees of morpho-transcriptomic correlation are observed among 11 additional cancer entities, for which PDL1 is currently not considered to be a relevant biomarker. These analyses reveal that the molecular basis of tumors can be depicted from the view of cellular morphology via advanced deep learning techniques, which

could provide a perspective on studies of tumorigenesis and treatment.

## Results

### Workflow of MILTS

The corresponding workflow of MILTS is presented in Fig. 1a. In the context of MILTS, a slide-level label initialized from dichotomized mRNA expression levels is employed to supervise the representation learning for the histopathological image, where three binarization thresholds (quartile, tertile, and median points) for each cancer type were considered. Specific values for each cutoff can be found in Supplementary Tables S1 and S2. Because the slide-level labels constitute an aggregate summary, which is expected to differ across the tiles of the tumor section, the teacher-student framework combines dynamic label assignment for individual tiles with knowledge distillation from a temporal ensemble model representing the exponentially decaying average of previous learning iterations. Specifically, tiles are processed both the teacher and student models with random augmentation including rotation, crop, flip and color transformation. The teacher model continuously yields tile-level pseudo-labels for typical positive/negative tiles in each training epoch, based on which the student model is updated following the MIL constraint as well as the distribution generated by the teacher model on unlabeled instances. As the teacher model is continuously updated via the moving average of the student model, this collaborative procedure automatically learns tile level labels. These features are further fused by a transformer to obtain a slide-level token and combined with statistical summaries of patch-level outcomes in a multi-layer perceptron (MLP) to infer per patient results. More details are provided in the Methods section.

### Predictability of *PDL1* expression across nine cancer types

The ability to classify gene expression was evaluated on nine cancers for which PDL1 expression serves as an established biomarker for checkpoint inhibitors. The distribution of slides is shown in Fig. 1b (involving 3121 cases with 4215 fresh-frozen slides and 2966 FFPE slides). Using the upper tertile of *PDL1* expression as a threshold, MILTS achieved performance on fresh-frozen slides with a weighted average average area under receiver operating characteristic curve (AUC) of 0.83 (range: 0.64–0.91), accuracy of 0.75 (range: 0.58–0.87), sensitivity of 0.83 (range: 0.74–0.90) and specificity of 0.71 (range: 0.47–0.89). We also evaluated the model using other two threshold settings: the upper quartile (top 75%) and median (50%) expression levels in each cancer type. These two alternative thresholds yielded broadly comparable performance with a mean AUC of 0.81 and 0.75, respectively (Supplementary Fig. S3 and Supplementary Tables S3 and S4). In the following, we discuss results at the upper tertile, unless stated otherwise.

Specifically, the dataset comprises bladder urothelial carcinoma (BLCA), cervical squamous cell and endocervical adenocarcinoma (CESC), colon adenocarcinoma (COAD), kidney renal papillary cell carcinoma (KIRP), lung adenocarcinoma (LUAD), lung squamous cell carcinoma (LUSC), skin cutaneous melanoma (SKCM), stomach adenocarcinoma (STAD), and triple-negative breast carcinoma (TNBC). Performance results are shown in Fig. 1c. The threshold to determine the accuracy, sensitivity and specificity is selected by Youden's J statistic. The optimal threshold here is the one that maximizes the sum of sensitivity and specificity. This strategy is applied across all our experiments unless stated otherwise. 95% confidence interval (CI) is also computed with bootstrapping strategy (2000 random resamples), where the AUC performances on fresh-frozen slides are respectively 0.87 for bladder urothelial carcinoma (95% CI: 0.84–0.90), 0.91 for cervical squamous cell and endocervical adenocarcinoma (95% CI: 0.88–0.93), 0.92 for colon adenocarcinoma (95% CI: 0.90-0.93), 0.77 for kidney renal papillary cell carcinoma (95% CI: 0.73–0.80), 0.74 for lung adenocarcinoma (95% CI: 0.72–0.77), 0.73 for lung squamous cell

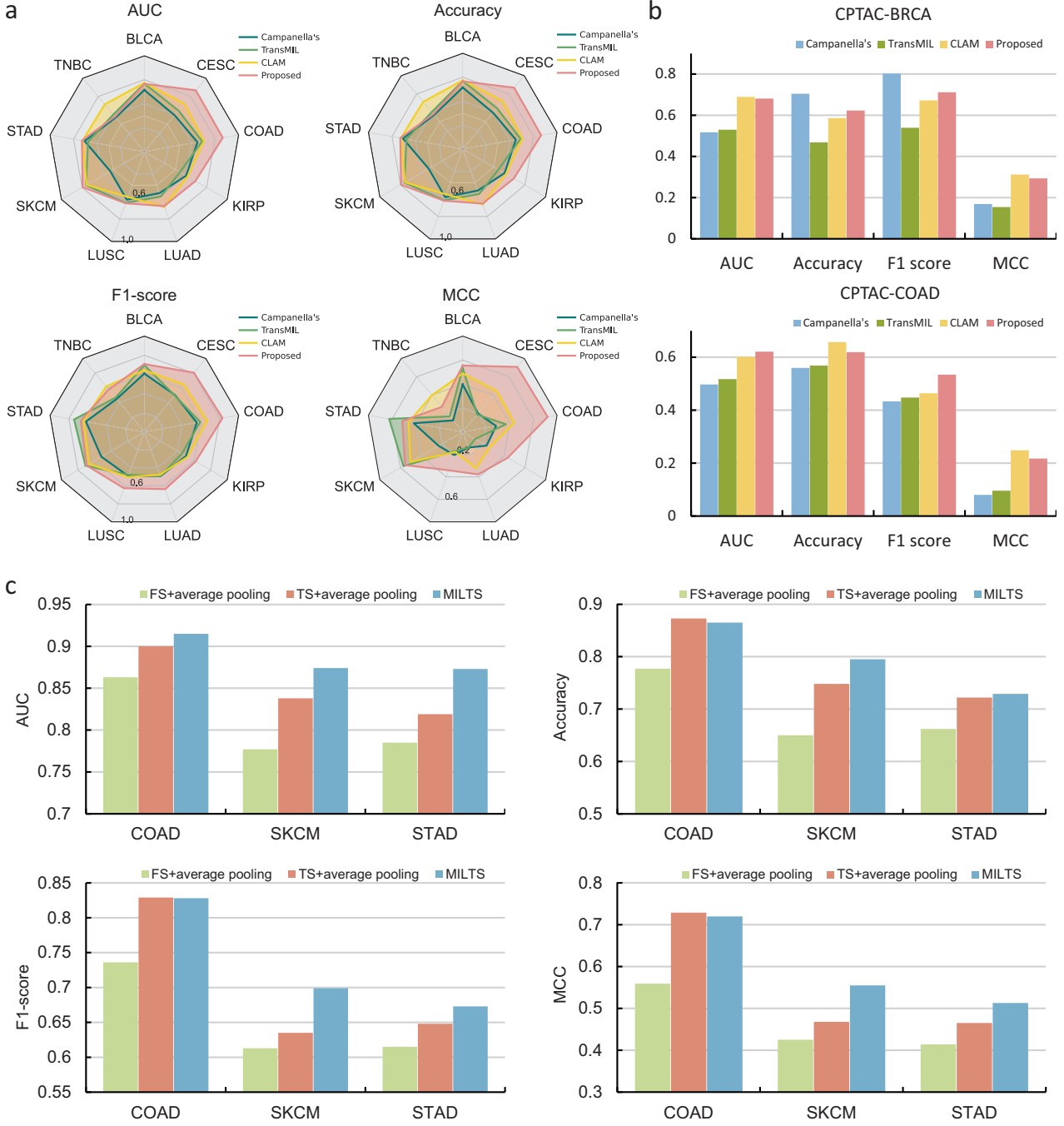

**Fig. 2 | Quantitative comparison of *PDL1* expression in clinically relevant tumors with other methods and ablation study results. a** Radar charts, arranged from left to right and top to bottom, represent the AUC, accuracy, F1 score, and MCC of the proposed method and comparison methods, respectively, at the tertile threshold. **b** The histogram shows the results of external validation on the CPTAC BRCA and COAD datasets using the same thresholds as in the TCGA datasets.

**c** Histograms of ablation study with respect to AUC, accuracy, F1 score and MCC. In the group of "FS + average pooling", a fully supervised framework and average pooling of patch-level predictions was adopted. In the group of "TS + average pooling", the patch-level feature aggregation module of MILTS was substituted with average pooling. Source data are provided as a Source Data file.

carcinoma (95% CI: 0.70–0.76), 0.87 for skin cutaneous melanoma (95% CI: 0.86–0.89), 0.85 for stomach adenocarcinoma (95% CI: 0.83–0.87) and 0.64 for triple-negative breast carcinoma (95% CI: 0.60–0.69).

Separate models were trained on FFPE samples using the tertile threshold. We maintained consistent cohort splits employed for fresh-frozen sections and an average AUC of 0.74 was achieved with the model trained with FFPE slides, where the trend in tumor-specific performance was consistent with that of fresh-frozen slides. Detailed results

are presented in Supplementary Tables S5 and S6. Nonetheless, there remained a performance gap between FFPE slides and fresh-frozen ones as shown in Fig. 1c. The finding that frozen slides usually yield better molecular inference aligns with observations reported in several previous studies[26,32]. Further investigations are warranted to explore strategies for bridging the gap between these two modalities. Overall, a significant morpho-transcriptomic link is evident regarding *PDL1* expression across nine PDL1-relevant cancer types, which demonstrates good predictability independent of the specific threshold.

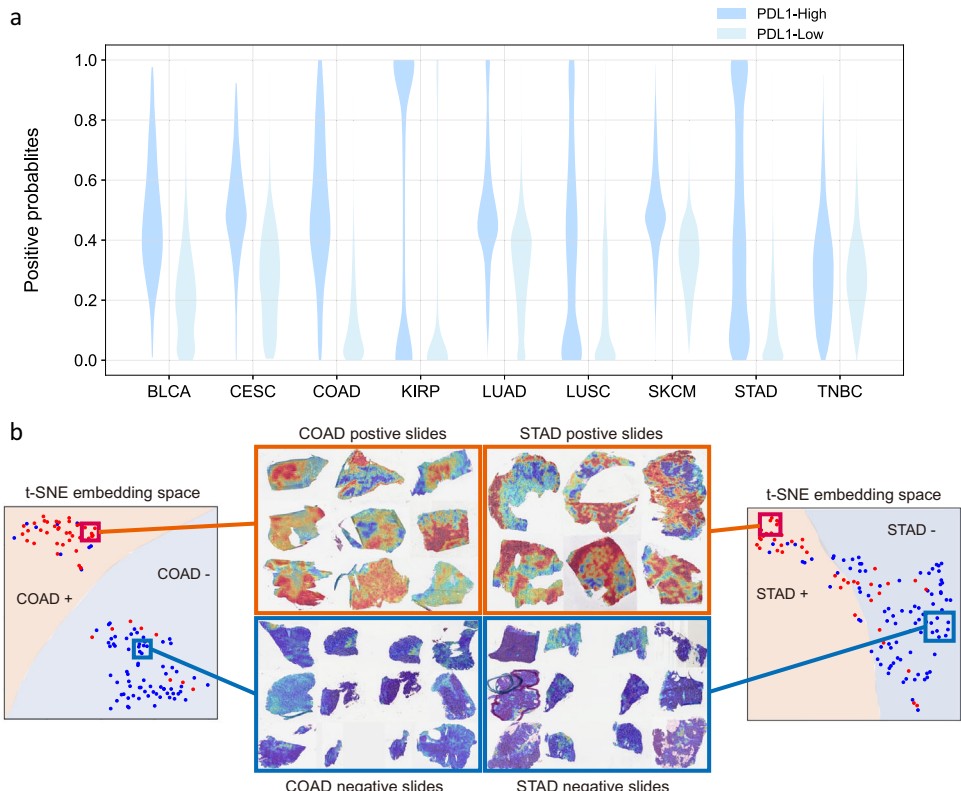

**Fig. 3 | Different distributions between samples of *PDL1* high and low expression. a** Violin plot of positive probabilities for tiles from *PDL1* high class and low class at thresholds of upper tertile for each kind of tumor. **b** Two-dimensional feature space constructed by t-SNE using 523-dimensional slide-level embedding.

Heatmaps shown in the red boxes are from positive samples and those in the blue boxes are from negative samples, within which red represents a high probability of being *PDL1* highly expressed and blue indicates the opposite. Source data are provided as a Source Data file.

## MILTS outperforms other methods in *PDL1* expression prediction

Comparison results demonstrate that MILTS outperforms other methods on aforementioned tumors, exhibiting a notable advantage of 9% or more in terms of average AUC. Evaluated were the method by Campanella et al.[36], TransMIL[46] and CLAM[47]. All comparison methods are deep learning-based MIL algorithms, among which method by Campanella et al. adopts instance-level strategy while TransMIL and CLAM work on embedding-level. The related hyperparameter settings are listed in Supplementary Table S7. Quantitative results are provided in Fig. 2a. The weighted average AUC of MILTS on these nine types of tumors is 0.83, while those of the method by Campanella et al., TransMIL and CLAM are respectively 0.63 (range: 0.51–0.75), 0.71 (range: 0.53–0.84) and 0.74 (range: 0.56–0.83). The results of F1 score and Matthews correlation coefficient (MCC) further reveal that MILTS exhibits more balanced sensitivity to both positive and negative samples, with an average performance of 0.69 and 0.50, respectively. In comparison, the second-best model achieves only 0.60 and 0.39 for the same metrics.

An ablation study was conducted to evaluate the individual modules of the teacher-student MIL learning and feature aggregation using COAD, SKCM and STAD data. Specifically, MILTS consists of two key modules: a teacher-student MIL module for patch-level feature extraction and a transformer-based feature aggregation module for slide-level predictions. The ablation study was conducted by examining these two modules in isolation. Details about the modules are provided in the Methods section. The results are presented in Fig. 2c. Results reported that teacher–student MIL module brought an improvement of 4.4%, 8.5%, 5%, and 8.8% with respect to AUC, accuracy, F1 score and MCC. The improvements of the feature aggregation module are respectively 3.5%, 1.5%, 2.9%, and 4.2%, respectively. Besides, we also implemented external validation on the CPTAC dataset[49] whose results are shown in Fig. 2b. The proposed method still demonstrated superior overall performance when compared to the other methods. However, all algorithms exhibit a decrease in performance when applied to the CPTAC dataset. This decline in accuracy could be attributed to several factors, including variations in mRNA quantification and differences in the slide image modalities used in the CPTAC dataset which are a mixture of fresh frozen slides and FFPE slides. Further investigation and refinement of the algorithms may be necessary to address these issues and improve their performance on the CPTAC dataset. Nonetheless, the quantitative analysis of comparison confirms that the proposed model appear to better decipher morphological features associated with *PDL1* expression from pathological patterns compared to other methods.

## Spatially heterogeneous patterns of high *PDL1* expression

Like most other MIL models, MILTS calculates predictions for each individual tile as well as for the entire slide. Strikingly, tiles from samples within the upper tertile of bulk *PDL1* expression are predicted to exhibit a wide range of *PDL1* expression, whereas tiles from the remaining samples were uniformly low (Fig. 3a). These distributions of tile-level probability accord with the assumption that there exists a substantial part of instances in positive slides which may not exhibit the same characteristics as the overall slide-level label suggests. Similar results were observed for the upper quartile and median thresholds, where it appears that the range of positive probabilities for the *PDL1*-low class tends to shrink as the threshold value increases (Fig. S4). Conversely the range of positive probabilities becomes wider when the median is chosen as the threshold, indicating that the model begins to

unravel nuances of very low *PDL1* expression. The distinct distributions between *PDL1* high and low groups explain the observed differences of AUCs (Figs. S5 and S6). Noting that the distributions show greater overlap of other thresholds reflects the lower AUCs for these cutoffs and indicates that the histopathological differences are less pronounced for median and upper quartile threshold.

The notion that positive tiles reflect common histological features is also supported by a UMAP[50] visualization of the embeddings learned by the patch-level classifier in Fig. S7. Here the representations of instances in slides with high *PDL1* expression exhibited varying degrees of overlap with instances from negative slides, and those with high expression also demonstrated their own distinct distribution, independent from the distribution of negative samples, which were also reflected respectively by the cold region and highlighted region in heatmaps of PDL1 positive slides (Fig. 3b).

### Deep morphological features help discovering morphotypes for *PDL1* expression

The aforementioned dispersion of high *PDL1* predictions across tiles from the same slide manifest in a spatially organized fashion as evidenced by heatmaps of the predicted *PDL1* expression (Fig. 3b). In the case of negative slides, the distribution tends to have a relatively lower mean value, resulting in the domination of blue regions in the corresponding heatmaps. In contrast, the heatmaps of positive slides exhibit different contiguous areas of predicted high and low expression values. This spatial heterogeneity also coincides with a range of histopathological patterns, which we illustrate at the example of colon adenocarcinoma.

Presented in the first row of Fig. 4a, it is common to observe a mixed inflammatory stroma characterized by a relatively high abundance of eosinophils in areas with high *PDL1* expression, which can be either intact or degranulated. This suggests an immune response or inflammatory process occurring in these areas. The findings in Ref. 51 align with our observations, where *PDL1* expression was predominantly observed on tumor-associated inflammatory cells by pairwise comparison with IHC slides in MSI-H subtypes. Another distinctive feature observed in areas with high *PDL1* expression is the presence of a cribriform (sieve-like) growth pattern of tumor cells in the second row of Fig. 4a. This growth pattern is accompanied by tumor cells exhibiting hyperchromatic nuclei, which appear darker and more intensely stained compared to surrounding cells. The presence of cribriform growth has been identified as an independent prognostic factor in various types of cancers, indicating a higher risk of tumor progression, metastasis, and decreased overall survival. In contrast, typical negative patterns usually included areas of non-invasive adenomatous parts of the lesion as illustrated in the first row of Fig. 4b. The non-invasive adenomatous areas appear more uniform and well-differentiated glandular architecture, indicating a lower likelihood of malignancy or aggressive behavior. Other patterns also include normal colonic crypts adjacent to the invasive carcinomas, tumor necrosis, abundant tumor-associated mucin in case of mucinous carcinomas and coagulation necrosis at the sample margin in Fig. 4b. The aforementioned recurring patterns were identified in both fresh-frozen and FFPE slides. Additional examples can be found in Supplementary Figs. S8 and S9.

Further, the predicted patterns of *PDL1* gene expression were also validated using paired PDL1 IHC slides. Correlation analysis between model predictions and IHC scores was conducted using a set of 20 colon adenocarcinoma samples. Visually, PDL1 IHC levels exhibited similar patterns as H&E based predictions (Fig. 5a). IHC levels were quantified in patches of $128 \times 128 \, \mu m^2$ and compared to the H&E based *PDL1* prediction in matching areas containing 1% of the total patches on the slide. Model predictions exhibit a consistent strong positive correlation with IHC across all 20 slides, with an average Pearson's correlation coefficient of 0.74 (Fig. 5b). Together these findings

confirm the model's ability to deconvolve gene expression signals and attribute these signals to distinct histopathological areas. More details are provided in the Supplementary Note 1.

### Correlations between MILTS predictions with TME and clinical features

In order to better understand the observed histopathological associations, correlations with the immune microenvironment, as estimated from RNA-seq data by CIBERSORT[52], and with other clinical parameters were performed. The analysis of immune infiltrates revealed an overall positive correlation between the presence of cell types, such as M1 macrophages and CD8+ cytotoxic T cells, and predicted *PDL1* expression across most cancer types (Fig. 6a). This finding aligns with the observed patterns of high *PDL1* expression in COAD which was found to co-occur with a mixed inflammatory infiltrate. The analysis also revealed a correlation between the presence of CD4 memory-activated T cells and elevated *PDL1* expression, which may indicate a previous immune response against tumor antigens, leading to the upregulation of *PDL1* as a countermeasure by tumor cells to suppress T cell activity and evade immune attack. This correlation may imply an intricate interplay between *PDL1* expression, immune cell infiltration, and the inflammatory response within the tumor microenvironment.

In addition, we conducted an analysis to assess the correlation between the predictions generated by MILTS and various clinical features specific to different cancer types which is shown in Fig. 6b. Across cancer types, only few consistent trends were observed, with tumor mutation burden and TIL Regional Fraction exhibiting generally weak positive correlations with *PDL1* prediction patterns. The finding of *PDL1* expression coinciding with high inflammation, which is often found in tumors with high mutation burden, agrees with the observations of the previous section. To further verify the correlation, we crafted three distinct feature sets, consisting of (i) the deep features derived from the proposed model, (ii) deep features concatenated with standardized clinical features and (iii) clinical features alone. An XGBoost classifier was employed to conduct the prediction based on the forementioned embeddings. The results are presented in Fig. 6c. It's evident that the deep features extracted by MILTS outperform the purely clinical features which implies that the proposed model encapsulates unique morphological information sourced directly from the pathological slides. We also implemented SHAP to showcase the relative importance of the deep features in comparison to clinical features (Fig 6d). Using SHAP values, it could be observed that the deep features derived from the proposed model hold significantly greater importance than the clinical features. These evidences indicate that the features extracted by the model are mostly either independent of, or lowly correlated with, standard clinical features. Consequently, it suggests that the features captured by the model are distinct and do not overlap with or duplicate the information provided by the clinical features. This highlights the complementary nature of the model's predictive capabilities in relation to the clinical characteristics of the cancers under investigation.

### Weaker histopathological associations in other cancer types

In order to provide broader context, we also conducted analysis of tumor types, for which immunotherapies have not been approved with PDL1 companion tests, or for which the prognostic significance of *PDL1* expression has not been significantly verified. The experiments involved a total of 11 types of tumors, which were adrenocortical carcinoma (ACC), esophageal carcinoma (ESCA), head&neck squamous cell carcinoma (HNSC), liver hepatocellular carcinoma (LIHC), mesothelioma (MESO), ovarian serous cystadenocarcinoma (OV), prostate adenocarcinoma (PRAD), rectum adenocarcinoma (READ), testicular germ cell tumors (TCGT), thyroid carcinoma (THCA) and uterine corpus endometrial carcinoma (UCEC). The statistical performance of

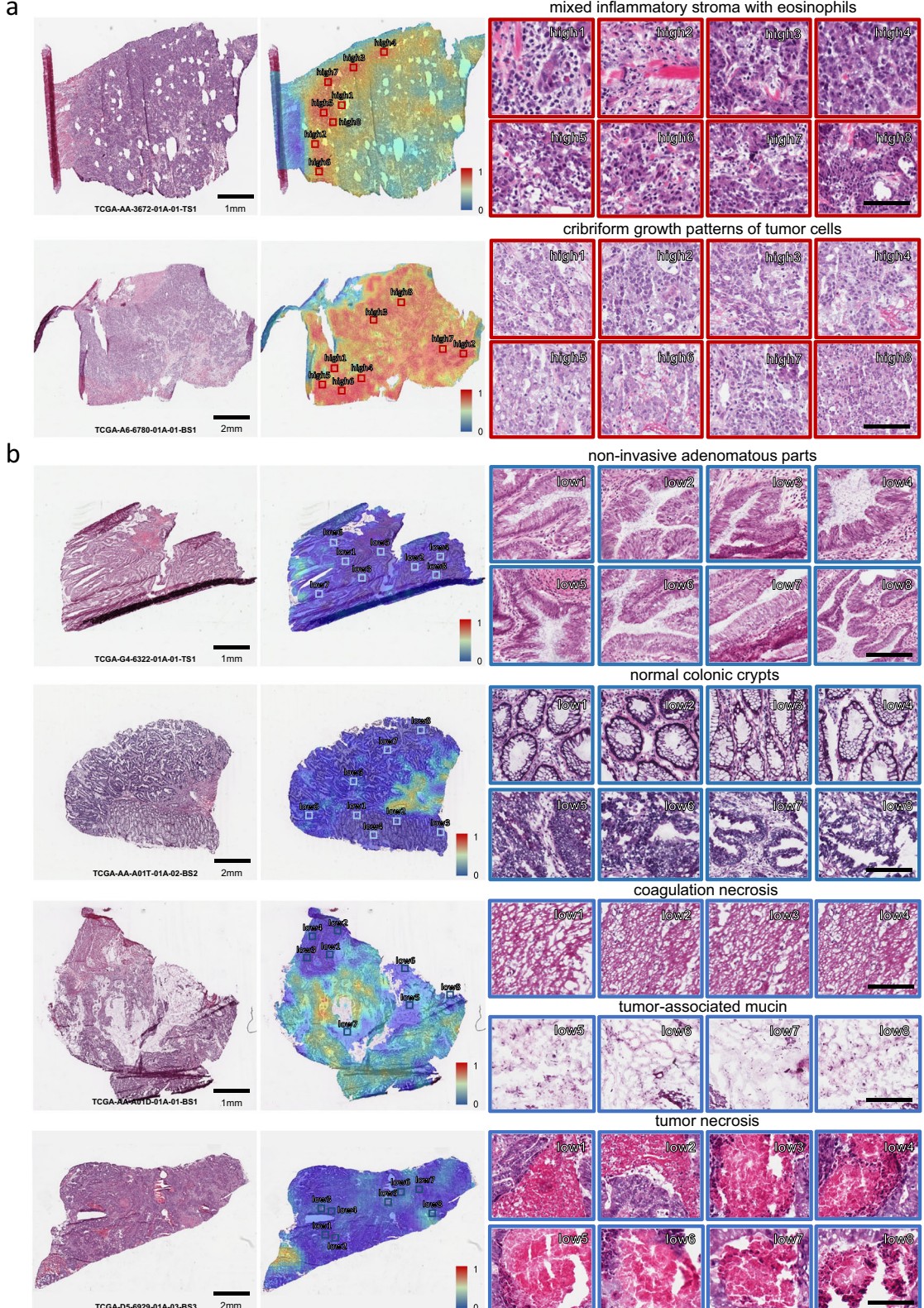

**Fig. 4 | Typical patterns for *PDL1* high/low expression in H&E slide images of COAD. a** Example slides with typical *PDL1* positive and (**b**) negative patterns. From left to right are the original H&E slides, predicted heatmaps and example tiles with high and low predicted *PDL1* expressions. Tiles of high expression are marked in the color red and low ones in blue. Scale bars in the tile views of (**a**) and (**b**): 100 μm.

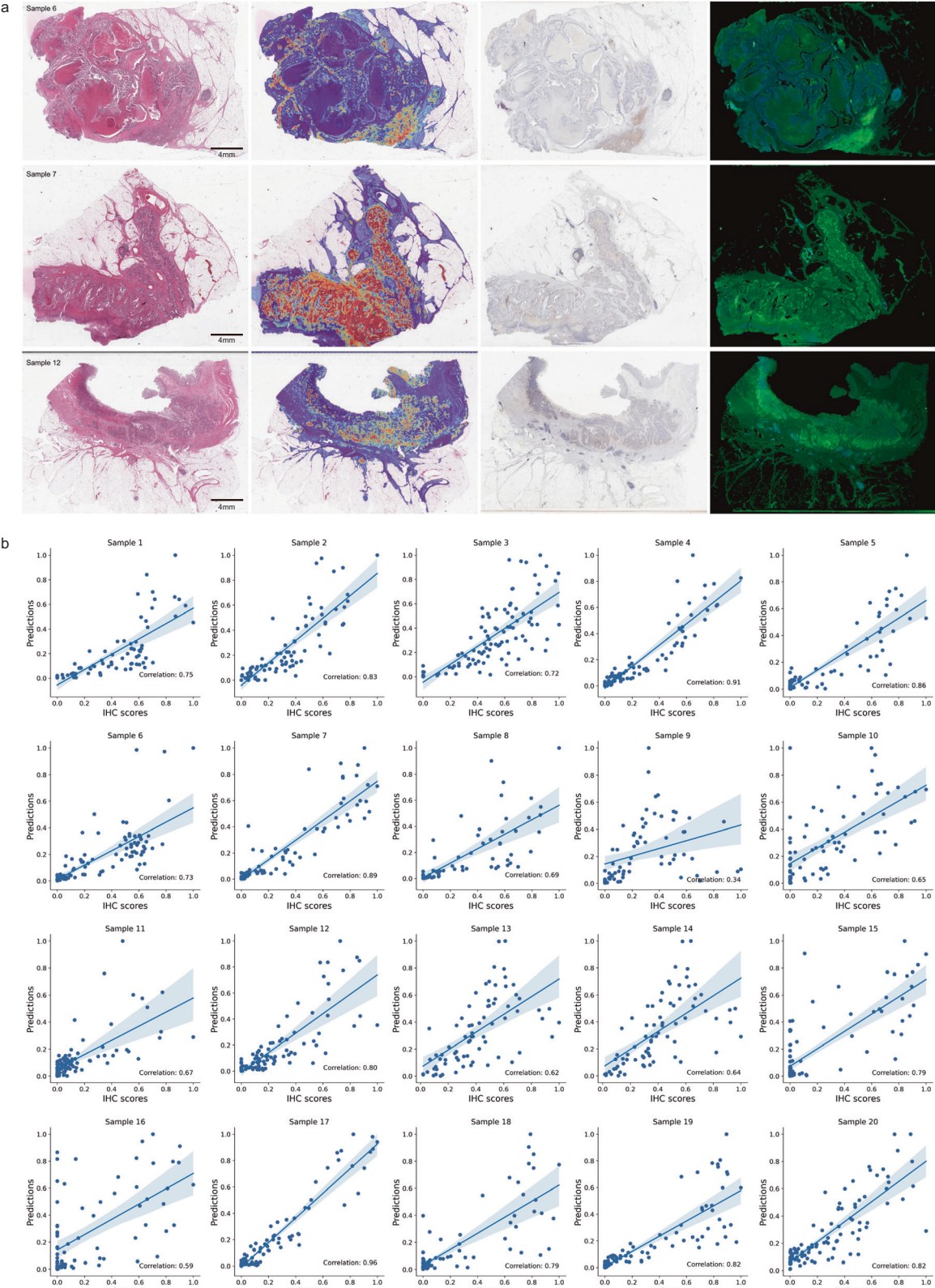

**Fig. 5 | Correlation analysis between model predictions and paired IHC quantification. a** Visual comparison between predicted heatmaps and corresponding IHC slide images. The stain-separated images are produced by employing the diaminobenzidine and hematoxylin channels from IHC slide images as the green and blue components, respectively. A more pronounced green area signifies higher PDL1 levels. **b** Scatter plots illustrating the relationship between normalized IHC quantification and predicted positive probability by the proposed model. The error band represents a 95% confidence interval, calculated using bootstrap methods. Source data are provided as a Source Data file.

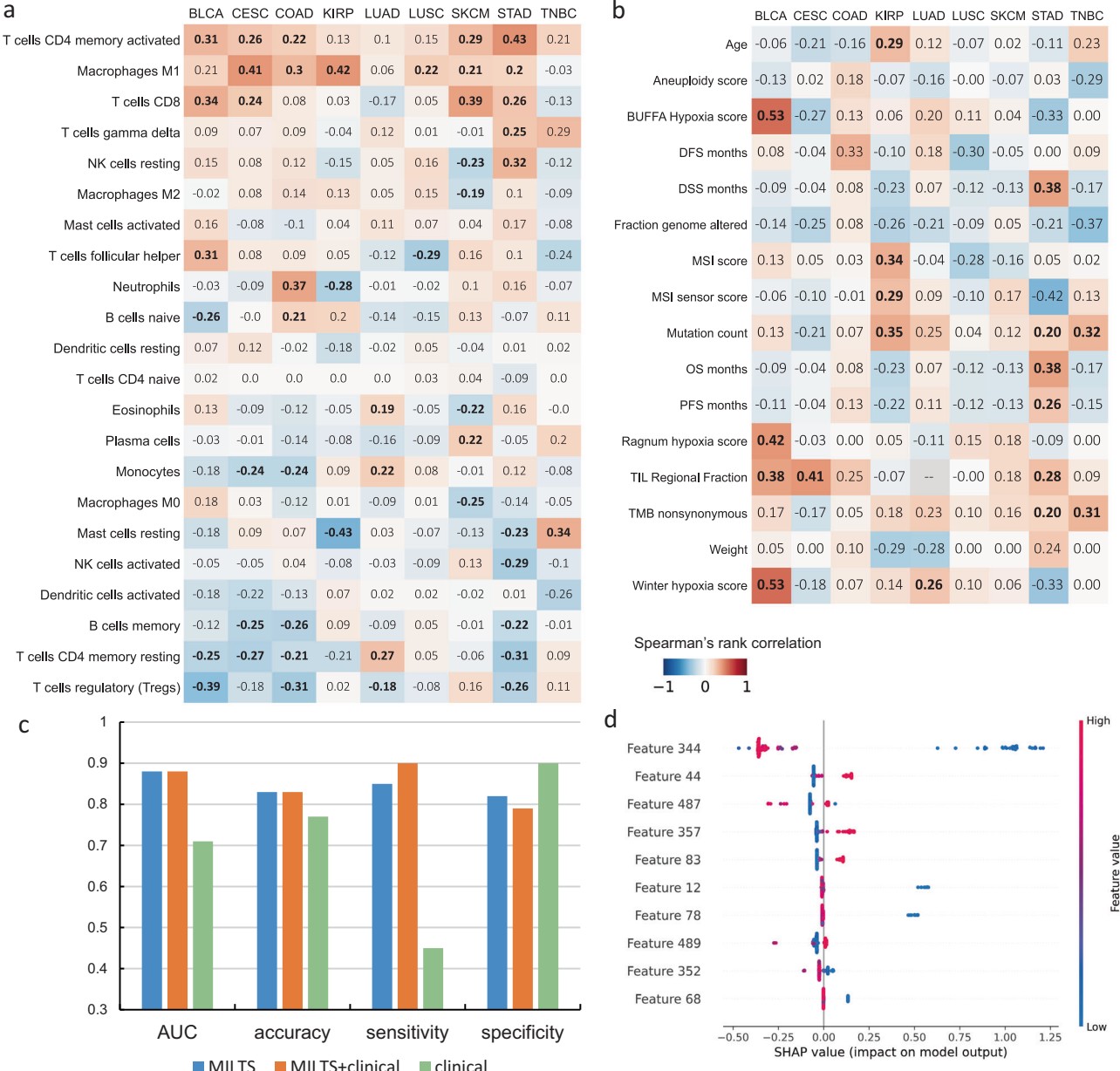

**Fig. 6 | Correlation analysis on tumor immune microenvironment and clinical profiles. a** Heatmap of Spearman's rank correlation between model predictions and immune infiltrates. **b** Heatmap showing Spearman's rank correlation between model predictions and clinical features. The bold values indicate correlation coefficients with *p*-values less than 0.05. The alternative hypothesis is specified as two-sided. **c** Classification performance by the deep features, deep features concatenated with standardized clinical features and clinical features alone. **d** Features with the top 10 SHAP values. Features with indices less than or equal to 523 represent deep features extracted by MILTS. Source data are provided as a Source Data file.

MILTS on these tumors is presented in Fig. 7a. The average AUC value is measured at 0.67 and the average accuracy stands at 64%, indicating a moderate level of discriminatory power in distinguishing between different mRNA expression levels. See Supplementary Tables S8–S10 for the details.

Among the 11 cancer entities analyzed, it is important to note that not all tumors display a distinct morphological pattern of *PDL1* expression that can be directly comparable to PDL1 relevant tumors. The AUCs for HNSC, READ, TGCT and THCA exceeded 0.7, which were comparable to the performance on cancers with PDL1 as an established biomarker where TNBC is the only one with the AUC lower than 0.7. Besides, we also devised a similarity measure based on the distribution of average classification performance of all PDL1 relevant tumors (see Similarity Measure in Supplementary Note 2). The corresponding results are exhibited in Fig. S11, and the four aforementioned tumor types (HNSC, READ, TGCT, and THCA) still demonstrate good similarity. This implies that the intermediate processes connecting macroscopic changes in tumor morphology to microscopic gene expressions may share certain common features for these tumors with good predictability of PDL1 expression, which contribute to their consistent and accurate prediction of *PDL1* expression.

*PDL1* expression was found to be lower compared to PDL1 relevant tumors, with mean values for the two groups being 1.71 and 3.27. Despite this positive correlation exists between the average level of *PDL1* expression and corresponding classification performance we note that there are some PDL1 relevant tumours which achieve high classification accuracy despite low *PDL1* expression (Fig. 7b). This implies that the distinction in the predictability based on

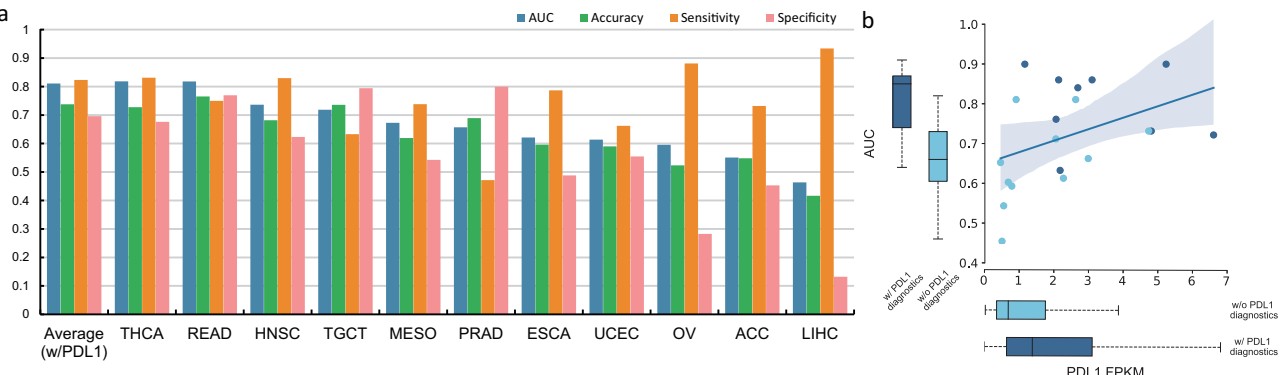

**Fig. 7 | Performance on other cancer entities. a** Histogram of model performance on the other 11 cancer entities indicated by AUC, accuracy, sensitivity and specificity. The average performance of the group with PDL1 diagnostics is displayed in the leftmost column. w/PDL1 means the tumors with PDL1 as an established biomarker. **b** Scatter plot showing the correlation between the overall *PDL1* expression level and AUC performance. Box plots displayed alongside the Y and X axes, represent the distribution of tumour-specific AUCs and the *PDL1* FPKM, respectively. The central line in each box represents the median value. The box spans the interquartile range (IQR), with its lower and upper boundaries marking the 25th and 75th percentiles, respectively. Whiskers represent the maximum and minimum values. In the AUC box plots, the groups are divided into those with established PDL1 diagnostics ($n = 9$) and those without ($n = 11$). For the *PDL1* FPKM box plots, the sample sizes are $n = 3830$ and $n = 3386$ for the two groups, respectively. Groups with and without PDL1 diagnostics are distinguished by dark blue and light blue markers in the data. w/, with; w/o, without. The error band represents a 95% confidence interval, calculated using bootstrap methods. Source data are provided as a Source Data file.

histopathology may be attributed to multiple factors in addition to *PDL1* expression level, which may also have implications for therapeutic outcomes. The residual differences in morpho-transcriptomic associations suggests that additional investigation is warranted for tumors with better morphological correlation.

## Discussion

PDL1 has been proven to be an effective biomarker for determining the response to the immunotherapies in a number of cancer types. Here, we developed MILTS, a weakly-supervised multiple-instance learning algorithm for predicting slide-level labels from H&E slide scans. We demonstrated its efficacy in predicting elevated *PDL1* expression for a broad range of cancers including 9 cancers where PDL1 serves as an established biomarker. Given the absence of a clinically optimal threshold, different cutoffs thresholds of mRNA quantification were employed in this study to verify the correlation. Compared to existing tools MILTS displayed competitive performance. Our analysis revealed salient links between histopathology and the level of *PDL1* gene expression for these tumors across various thresholds. Of note, tumor types for which PDL1 is not considered to be a relevant biomarker, also exhibited lower histopathological predictability, likely because of overall lower *PDL1* expression.

In addition to the predictability of *PDL1* based solely on histology, which could potentially help reduce the need for additional tests, this study also highlighted a diversity of histopathological patterns associated with high and low *PDL1* expression. Utilizing the deep histopathological features and patch-level predictions, typical patterns for high PDL1 expression in colon adenocarcinomas were found. These include a mixed inflammatory stroma characterized by a relatively high abundance of eosinophils and a cribriform growth pattern of tumor cells. Furthermore, low expression patterns include tumor necrosis, coagulation necrosis at the sample margin, tumor-associated mucin in case of mucinous carcinomas, normal colonic crypts adjacent to the invasive carcinomas, and, most interestingly, non-invasive adenomatous parts of the lesion. It reflected that this weakly-supervised learning manner was capable of pinpointing the potential morphological patterns or cytomorphology of different PDL1 expression levels, which due to their diverse nature and heterogeneous occurrence across and within a slide are difficult to establish by conventional means.

Despite these insights, there are still certain limitations. The first one is the persisting performance gap between fresh-frozen and FFPE

tissue sections in the proposed model. This discrepancy is likely due to the fact that fresh-frozen slides are generally considered to better preserve the structure of molecular content, as noted by ref. 26. Therefore, future research should focus on bridging this gap and enhancing the generalizability of models trained with single-modality data. This improvement is crucial for making AI models applicable and effective in routine clinical settings. Another important limitation was the consistency scale used for *PDL1* expression. The accuracy and reliability of the results heavily depended on the thresholding method employed to generate labels from mRNA expression levels. The potential mismatch between the bulk RNA sequences and the tissue presented in the histopathology slide can introduce challenges in integrating the molecular information from bulk RNA sequencing with the spatial information provided by histopathology slides. And variations of the protocols and techniques used for RNA expression quantification can also impact the knowledge transferability between different datasets. These may be plausible factors contributing to the observed performance differences in the external validation on the CPTAC dataset in this study.

Overall, our study revealed that it is possible to predict high and low *PDL1* mRNA expression based on H&E slides with the proposed weakly supervised method MILTS. Furthermore, MILTS captures and identifies meaningful histopathological patterns that are associated with molecular changes, which may not be readily discernible due to their heterogeneous occurrence across and within slides. These findings underscore the utility of AI to augment the capabilities of pathologists by leveraging large amounts of digitized histopathological slides with slide-level annotations, and thereby providing valuable insights into the complex interactions between histology and molecular biology.

## Methods
### Datasets
This study was approved by the Ethics Committee of the Charité University Medicine (#EA4/046/21) and complied with all relevant ethical regulations. Whole slide images and transcriptome profiling data utilized in the experiments come from TCGA project via National Cancer Institute (NCI) Genomic Data Commons Portal[53] and CPTAC project via the Cancer Imaging Archive (TCIA) Pathology Portal, which comprises 12,299 H&E stained whole slide images and corresponding mRNA quantification data obtained from 6715 patients diagnosed across 20

different types of cancer. Twenty paired FFPE and IHC colorectal cancer samples are obtained from Charité-Universitätsmedizin Berlin. The age, sex and other metadata may not be published due to regulatory reasons. Furthermore, colon cancer demonstrates no substantial sex-specific characteristics, and PDL1, a widely recognized biomarker, is established as independent of both sex and age factors. Thus, these metadata were not utilized in producing the research results of this paper, making them irrelevant for data interpretation and the reproducibility of results. Informed consent from all participants was obtained and human participants did not receive financial or any other compensation. Data from CPTAC dataset is employed for external validation purposes. The sample types comprise mainly primary solid tumor as well as metastatic tissues. Specifically, the dataset comprises 8719 fresh-frozen slides and 2966 FFPE slides from TCGA, along with 614 slides from CPTAC COAD and BRCA. Characteristics of patients are presented in Supplementary Table S11. We collect all available whole slide images scanned at a magnification of 20× or higher, along with their CD274 mRNA expression level read counts normalized by the upper quartile fragments per kilobase of transcript per million mapped reads (FPKM-UQ).

According to the biospecimen information available for TCGA cases, it is confirmed that the digitized fresh-frozen slides and sequencing data usually originate from the same vial of the same sample, with the slides typically obtained from either the top or bottom layer of the corresponding section. However, FFPE slides are sampled from a different vial and the potential mismatch with the sequencing data could be more pronounced. Thus a thresholding strategy was implemented on sequencing data converting the task into a classification task rather than a regression one, thereby enhancing the alignment between the data and labels. A commonly used split for data, in which 60% is reserved for training, 15% for validation, and 25% for testing, was employed for the majority of the tumors analyzed. It's worth noting that the splitting was performed based on patient IDs. This led to a distribution of 5373 slides for training, 1189 slides for validation, and 3182 slides for testing in the case of fresh-frozen slides, and 1837, 386, and 743 slides respectively for FFPE slides.

## Data preprocessing

Due to the tremendously large size, whole slide images should be decomposed into smaller elements so that deep learning techniques like convolutional neural network could handle them at an acceptable computational cost. Hysteresis thresholding is applied to exclude the invalid background by finding contours of stained tissues. The valid regions within the contours or partly overlapped are cropped into tiles of 256 pixels × 256 pixels at resolution of 20×, which correspond to the physical size of 128 µm × 128 µm respectively. In total, 52,587,256 tiles are extracted for the establishment and verification of the model. To mitigate the bias introduced by variations in staining protocols and laboratory conditions, we applied random adjustments to the brightness, contrast, saturation, and hue of generated tiles. In addition, random rotation and crop of the tiles were also applied to increase the variability of scales and orientations. Detailed parameter settings for data augmentation are shown in Supplementary Table S12.

## Teacher-student collaborated multiple instance learning (MILTS) for patch feature extraction

A two-stage strategy is adopted in the diagnosis of a patient's slide, starting with patch-level assessment, followed by slide-level feature aggregation and prediction. This framework enables not only higher-level diagnosis of the patient, but also interpretation of the slide's inner structure based on the patch-level predictions. Specifically, prediction of *PDL1* expression is formulated as a classification problem using a specific mRNA quantification value as the threshold. This threshold was set based on the median point, upper tertile, and upper quartile of the mRNA quantification distribution. Classification based on the

patch-level classifier is usually devised under multiple instance learning framework when only the slide-level property is known and potential heterogeneity is implied. During the patch-level feature extraction stage, MILTS made two important improvements. Firstly, we modified the optimization of multiple instance learning framework to a class-targeted way considering the heterogeneous composition of a slide image. Secondly, inspired by the human-decision making process, the single model of MIL is decomposed into a teacher and a student model, in which the teacher model dynamically assigns probabilities to tiles, and the student model subsequently learns to predict. These two components work collaboratively to boost the performance on patch-level pathological feature extraction.

In the classic MIL approaches, the target label $Y_k$ is predicted based on the distribution of the instances (or patches) $\{x_{1,k}, x_{2,k}, \ldots, x_{n,k}\}$ from a bag (a slide image in this case) $X_k$, where the hypothetical distribution should follow the principle below[54]:

$$Y_k = \begin{cases} 0, & \text{if } \sum y_{i,k} = 0 \\ 1, & \text{else} \end{cases} \tag{1}$$

where $y_{i,k}$ is the label for instance $x_{i,k}$ and the principle applies to the binary classification. This assumption implies that positive instances should only be present in positive bags, which accurately reflects certain scenarios in pathological diagnosis, such as distinguishing between benign and malignant lesions.

However, it is crucial to note that PDL1 expression is quantified by the accumulation of cancer cell surface PD-L1, rather than simply by its presence or absence. This implies that elements defined as highly or lowly expressed could exist in cohorts with contrary macro statistical properties. Based on this observation, we propose our class-targeted optimization way, which regularizes the instance-level learning by filtrating the representative instances for slides with varying labels. Instances from different slides are selected based on class-representative criteria defined by their respective slide labels. Specifically, a maximum-minimum selection criteria is established to determine the groups of instances involved in model training for binary classification. In the scenario where patients should be classified as high or low *PDL1* expression, instances from positive samples with maximum response and negative samples with minimum response tend to be selected and share the same label with bags they belong to. The criteria could be summarized as:

$$\tilde{i}_k = \begin{cases} \arg\max_i (f_\theta(x_{i,k})), & Y_k = 1 \\ \arg\min_i (f_\theta(x_{i,k})), & Y_k = 0 \end{cases} \tag{2}$$

where $\tilde{i}_k$ is the index of the selected instance and $f_\theta$ is the function of instance-level classifier which maps the input patch $x_{i,k}$ into a normalized hidden variable indicative of its typicality for the slide. To incorporate additional histological pattern references, the criteria can be further relaxed by allowing $M > 1$ representative instances in the slide.

Building on our improved class-representative MIL approach, we further consider data composition in this task as a mixture of labeled and unlabeled instances and introduce the teacher-student collaborative learning into MIL according to the characteristics of whole slide images and weakly annotations. Typically, the representative instances selected from a slide are considered as labeled data and assigned the same label as the slide. Nevertheless, these instances only represent a small portion of the entire slide image. Relying solely on these instances for training can result in overfitting and suboptimal performance due to their limited representation of the entire slide image. There still exist a considerable number of instances within whole slide images that also contain pathological information and can be utilized for representation learning. Usually to address the

challenge of interpreting slides containing both clear and ambiguous regions, a pathologist typically relies on their own expertise to interpret the clear parts of the slide (labeled) and consults with more experienced or senior pathologists for guidance on the ambiguous regions (unlabeled). And this process also benefits senior pathologists by exposing them to difficult cases. Taking inspiration from this process, we propose to decompose the single model in the MIL framework into a temporal ensemble model consisting of a teacher model, acting as the senior pathologist retrieving regularization of unlabeled data on feature representation, and a student model, learning from the teacher model. The mean teacher method[55] is adopted to constructed the teacher model and ResNet34[56] as baseline architecture for teacher&student model. Specifically, labeled instances $X_{labeled} = \{x_{1,1}, x_{1,2}, .., x_{K,1}, .., x_{K,n}\}$ selected by the teacher model, where $K$ instances of each slide are assigned with the label consistent with the sample, and unlabeled instances $X_{unlabeled} = \complement_X X_{labeled}$ (i.e., the complement of $X_{labeled}$) are both inputted into student model and compute a probability of high $PDL1$ expression in every iteration of training. The weighted cross-entropy loss function, which is used as the classification cost, penalizes the difference between the output and pseudo label for the clear regions of the slide represented by $X_{labeled}$. Meanwhile, the consistency cost is applied to the ambiguous regions represented by $X_{unlabeled}$ and constrains the distance to the distribution given by the teacher model. The loss function was formulated as:

$$
\begin{aligned}
L &= L_{WCE}(X_{labeled}, \theta) + \lambda L_{cons}(X_{unlabeled}, \theta, \theta') \\
&= -\sum_{x_i \in X_{labeled}} (\omega_0 y_i log(f_\theta(x_i)) + \omega_1(1-y_i)(1-logf_\theta(x_i))) \\
&\quad + \lambda \sum_{x_j \in X_{unlabeled}} \left\| f_{\theta'}(T(x_j)) - f_\theta(T(x_j)) \right\|^2
\end{aligned}
\tag{3}
$$

where $\omega_i$ is the weight to balance the class frequency and $f_{\theta'}$ is the function of the teacher model generated by exponential moving average (EMA) of previous student models. $T()$ is a combination of transforms including random rotation and color jitter. The parameters of the teacher model would be updated after the student model is optimized in each iteration.

In other words, the proposed approach constructs a teacher model to assess the reliability of knowledge and pairing the reliable parts with corresponding answers for the student model to learn. For data with greater uncertainty judged by the teacher model, the student model will attempt to imitate the teacher model's response even if an exact answer is not available. The teacher model will then use feedback from the student model to update itself and improve its own knowledge. Among different epochs of training, labels are dynamically assigned for instances according to the predicted probability of the teacher model via the above mentioned MIL manner, where meaningful histopathological patterns are expected to be uncovered through aggregated information distilled from mean teacher. And the student model optimized by gradient descent with regard to objective function $L$ would promote the prediction of the teacher model in turn by a weighted average behavior over training steps. This iterative process allows for continuous learning and refinement of both the teacher and student models to capture the effective histopathological features associated with PDL1 expression. Usually both models could be utilized to output the positive probability at the end of the training which share similar performance for patch-level prediction.

### Enhancing patient-level diagnosis by fused slide tokens with statistical summary of patch-level predictions

In the slide-level feature aggregation stage, statistical summary of patch-level predictions are merged with slide tokens by leveraging the local details extracted by the patch-level CNN and the global perspective of a transformer model. Given the trained model $f_{\theta'}$ or $f_\theta$ at

patch level, the typicality of all effective patches in a slide would be computed, which could be further interpreted as positive or negative probability according to the slide-level property. Based on the patch-level prediction, several statistical features of the slide were extracted including percentage of positive patches, histogram of probability distribution, median value and mean value of positive probability. Specifically, patches with typicality over 0.8 and below 0.2 are removed when computing the above features because a trimmed estimator is supposed to better reflect the central tendency of data. Further, features $e = [\mathbf{e_1}, \mathbf{e_2}, ..., \mathbf{e_m}]$ extracted by the patch-level backbone (i.e., feature vectors output by the adaptive pooling layer of ResNet34 in the end) are aggregated to form the global representation of the slide through the attention mechanism. Transformer is utilized to fuse these features into a class token $\mathbf{H}_k^{(0)}$ which could be formulated as[46]:

$$
H_k = LN(MSA(e)) \tag{4}
$$

where $LN(\cdot)$ is layer normalization and $MSA(\cdot)$ indicates multi-head self attention operation. The first row vector of correlation matrix $\mathbf{H}_k^{(0)}$ i.e., class token is taken as the aggregation of all local features in a slide. Training details follow the settings in ref. 46. Along with the statistical features mentioned above, the representation of the slide $\mathbf{E} \in \mathbb{R}^{1 \times 523}$ was constructed by concatenation and conducted the diagnosis by using MLP to classify the slide representation as $PDL1$ high expression or low expression. The MLP model is trained with cross-entropy loss and SGD optimizer, where the learning rate is set as $2 \times 10^{-4}$.

### Implementation details

Data preprocessing mainly including segmentation and patching of whole slide images was implemented on multiple workstations considering massive amount of data, whose central processing units are respectively AMD Ryzen 5950X, Intel i9-12900K, and Intel i9-10900K. Our weakly supervised framework MILTS and patient-level feature aggregation&diagnosis models were then trained and evaluated on a platform of GIGABYTE GeForce RTX 3090Ti with 24 GB of graphics memory. Algorithms were mostly programmed with Python (version 3.8.10) and libraries mainly involved were OpenCV (version 4.5.3), OpenSlide (version 1.1.2), Pillow (version 8.4.0), scikit-image (version 0.18.3) and Numpy (version 1.19.5). We utilized Pytorch[57] as the basic machine learning framework for data loading, enhancement, training and inference pipeline. Before the start of MILTS training, we initialized ResNet34 with ImageNet[44] pretrained parameters for both teacher and student model. A mini-batch size of 512 was adopted to accelerate computation and we used a stochastic gradient descent (SGD) optimizer with an initial learning rate of $1 \times 10^{-2}$ to optimize weights of the models. The proportions of labeled instances within a single slide were defined as 0.25, 0.35, and 0.45 for the quartile, tertile, and median points, respectively. The consistency cost weight $\lambda$ for the loss function was determined to be 100. EMA decay $\alpha$ for updating the teacher model was set to 0.99 throughout training. The strategy of cosine annealing was applied to schedule the learning rate and the minimum value was set as $1 \times 10^{-4}$. All the models were trained for 30 epochs, during which two circles of learning rate change were completed. On our platform using a single GPU, the typical inference time for a 20× whole slide image with non-overlap patches was 6.96s, which indicated a high efficiency and may be further promoted by large-scale parallel computing.

### Reporting summary

Further information on research design is available in the Nature Portfolio Reporting Summary linked to this article.

## Data availability

The TCGA data (images, as well as transcriptomic and clinical data) used in this study are publically available from http://gdc.cancer.gov. CPTAC image data are publically available from https://wiki.cancerimagingarchive.net/display/Public/CPTAC+Imaging+Proteomics and transcriptomic data from http://gdc.cancer.gov. The imaging data from Charité-Universitätsmedizin Berlin is accessible upon request. The data will be shared only under the condition that the request is for non-profit, purely academic research purposes, and the requesting researchers must provide valid ethics approval from their institution. The data generated in this study for the creation of the figures are provided in the Source Data file. Source data are provided with this paper.

## Code availability

The code in this paper is available through a Code Ocean compute capsule (https://codeocean.com/capsule/9197393/tree/v1)[58].

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

## Acknowledgements

This work was supported by grants from the National Natural Science Foundation of China under Grant 62271016, in part by the Beijing Natural Science Foundation under Grant 4222007, in part by the National Key Research and Development Program of China under Grant 2019YFB1311301, in part by the Fundamental Research Funds for the Central Universities and in part by the scholarship program of the China Scholarship Council under Grant 202206020151. The laboratory of T.G.P.G. is supported by the Barbara and Wilfried Mohr Foundation, the Dr. Leopold and Carmen Ellinger foundation, the Matthias-Lackas foundation, and the European Research Council (ERC-2023-COG 101122595). M.G.'s research group is endowed by the Robert Bosch foundation. We would also like to extend our thanks to Areeba Patel, Yiheng Tang, Heng Luo, Domenico Calafato, Gleb Rukhovich, Artem Lomakin and Anna Mathioudaki for helpful discussions and assistance in data processing.

## Author contributions

D.J., M.G., and X.B. conceived the study. D.J. developed the deep learning algorithms. D.J., S.L., A.S. and T.G.P.G. performed the experiment analysis. T.G.P.G., D.H., and A.A. prepared and collected pathology slides. T.G.P.G. and D.J. conducted the histopathological analysis. D.J., M.G., and X.B. wrote the manuscript with the help of all other authors. M.G. and X.B. supervised the project.

## Funding

## Competing interests

The authors declare no competing interests.
