## [Peer Review File · Nature Communications]

REVIEWER COMMENTS

Reviewer #1 (Remarks to the Author): Expert in digital pathology, machine learning, deep learning, and immunotherapy biomarkers

Jin et al employ a novel extension of multiple instance learning wherein the probabilities of the individual tiles are optimized to predict slide-level labels of PD-L1 expression. Extensive comparisons are made to other deep learning based multiple instance learning algorithms. The study is impressively comprehensive, involving almost 10,000 slides across 20 tumor types. The figures are well done and clearly convey the results of the study. The histological correlations are intriguing and suggest new avenues of research.

However, there are a couple points that should be explored more carefully:

- The study was conducted on fresh-frozen tissue which is not clearly mentioned until later in the manuscript. It is well known that histological morphology in FFPE can differ greatly from fresh-frozen tissue, degrading the histological correlation conclusions of the study. The utility of the algorithm also becomes into question as it is not applied to a routine diagnostic modality. Would it be possible to better characterize how well or how much further training would be required to achieve high performance on FFPE slides?
- Was the sequencing performed on the same block of tissue that the digitized slide was made from? Could that confound the observed histological correlations?
- The study predicts PD-L1 expression as characterized by mRNA. While Pare et al suggests a relationship between PD1 mRNA abundance and response to therapy, they do not find a correlation to PD-L1 tumor expression via IHC: "Second, we investigated the correlation of PD1 mRNA with PDL1 IHC in 74 evaluable samples (63%) from the validation dataset. No correlation between the two biomarkers was found ($r = -0.04$)" If their finding is robust and PD1 mRNA abundance is a better biomarker of response to therapy, then the study should be re-worded to emphasize that the predictive task is based on an mRNA PD1 abundance target as it is anyway suggested to be better associated to outcomes than PD-L1 expression by IHC.

Minor points:

References should be provided to other PD-L1 IHC prediction studies including Shamai et al (<https://doi.org/10.1038/s41467-022-34275-9>) and Sha et al (https://doi.org/10.4103/jpi.jpi_24_19).

Could a different type of plot be used in Fig 1B? I realize there is already an abundance of bar plots in this figure but a horizontal bar chart may better convey the composition of the dataset.

Are there any other studies that link histological features with PD-L1 expression as a way of further interrogating your findings?

What was the purpose of characterizing PD-L1 expression in tumors where immunotherapy is not heavily used? Why was the usage of immunotherapy in these tumors correlated to the performance of the algorithm?

L64: It is inferred that IHC is comparably expensive to RT-qPCR which is not true.

L121: What is the threshold for dichotomization?

L178: Why was the ablation study limited to COAD, SKCM and STAD?

L242: The distinction between M1/M2 macrophages is not well defined in humans and the focus could instead be on the function rather than class.

Reviewer #2 (Remarks to the Author): Expert in machine learning, deep learning, digital pathology, and immunotherapy response prediction

The paper offers a novel and effective approach to deep learning on WSIs for bio-marker classification tasks in which bio-marker expression has a significant intra-slide variance. In clinical practice pathologists make diagnosis decisions only after finding many instances of bio-marker expression in the slide. No individual instance can indicate the general slide's bio-marker presence. The paper captures the diagnosis' cumulative and smooth nature by redefining the MIL objective as a local maximum-minimum selection criterion. The criterion is based on a global slide label, as well as the model's current confidence in the local label. Using the student-teacher paradigm to create local "pseudo-labels" is an interesting way to upweight the most impactful patches in each slide. Utilizing the machine-learning innovations for patch-level representation and a transformer encoder for slide-level aggregation yields non-trivial results for mRNA-based PDL1 classification in various H&E datasets.

The paper contains clinical findings that reveal links between the slides' morphological features and PDL1. The model performed much better in tumor types where PDL1 is known to have prognostic significance compared to tumor types where its prognostic significance is yet to be proven. Interestingly, upon closer examination, for some cancer types in the group with no proven prognostic significance, the model seems to perform correlatively to the cancers for which there is evidence of prognostic significance. This inter- and intra-group difference implies that tumors that are clinically sensitive to PDL1 blockers have a morphology that is sensitive to the presence of PDL1 and that, in some specific tumor

types (HNSC, MESO, TGCT), there is a morphological sensitivity to PDL1 that might indicate as-of-yet unfound prognostic significance.

Another interesting finding is the link between the models' localized labels and known morphological features that characterize PDL1-positive slides in colon cancer. This link increases the credibility of the link between mRNA estimation of PDL1 and pathological diagnosis based on morphological features.

The paper isn't free of weaknesses. The models ground truth for training and performance evaluation uses mRNA for predicting PDL1 expression. Use of mRNA in such a manner is not part of practical clinical protocols. In fact, IHC staining is the current method used for determining the biomarker's presence. Thus, immediate clinical implications are in question.

An important issue is the seemingly liberal changes in mRNA dichotomization during test time. It seems that at some point during the research, the model was tested on three different label quantization levels according to different quantiles of mRNA expression. Only after observing optimal results for the model was the tertile chosen as the dichotomization point. If this is true, it means there was substantial data leakage into the test data before any further experiments were conducted.

Nevertheless, owing to the innovations, results, and potential clinical implications mentioned above, the paper is compelling and offers unique insights both clinically and technologically. The paper can be substantially improved by addressing the above and following issues:

- 1) To ensure the validity of clinical conclusions, it is necessary to rerun all experiments displayed in Figures 1, 2, and 6 using the mRNA expression upper quantile and median point as the dichotomization point. This rerun is particularly crucial for the nine cancer types that lack proven prognostic significance for PDL1. The current inferior performance of the model on these cancers can be attributed to overfitting the original 11 cancer types caused by a data leak. Therefore, if the tertile was chosen as the dichotomization point after conducting experiments on test data, it is crucial to conduct this rerun.

- 2) Threshold-dependent statistics such as accuracy, sensitivity, specificity, and F1 can only be discussed if the reader understands the threshold for model classification. In line 151, Youden's J statistic is mentioned as the method for finding the threshold. Is this statistic methodology used for all experiments? If not, how were the thresholds chosen?

- 3) The ablation study mentioned in line 177 should come after some description of the model architecture, as the reader still needs to be made aware of the different modules contained in the model.

4) The supplementary material should mention specific parameters used for model augmentation.

5) Neither the hyperparameters used for training MILTS nor the method by which they are found are mentioned in the paper. The hyperparameters used for comparison with the competing methods (Campella's, CLAM, TransMIL) are included in Table S3, but there is no mention of how these were found. For the comparisons in Figures 2a and 2b to be convincing, the reader needs to be convinced that all models compared had a fair chance.

6) Beginning in line 374, the sentence "This threshold was set based on the median point, upper tertile, and upper quartile of the mRNA quantification distribution" is the only mention of the dichotomization of mRNA used during training time. The paper lacks an exact explanation of how the threshold was chosen in training time.

7) The sentence beginning in line 356, "A commonly used split for data, in which 60% is reserved for training, 15% for validation, and 25% for testing, was employed for the majority of the tumors analyzed." raises questions as to the splitting strategy. The paper needs to mention whether slides taken from patients included in the training and validation sets were also found in the test set. If so, the split may cause data leakage from the test set and raise concerns about unmeaningful results. One quick and imperfect way to remedy this split problem is to remove patients in the test set that also appear in the training set. If such a data contamination exists, there is a need to redo the split properly and re-run all experiments.

8) The claims made using Figure 5b are unconvincing. A simple rank correlation between 2 features does not support the claim that the features added unique data previously unavailable. One way to show this claim is accurate is to try to fit a model on the slide representation token, concatenated with the clinically significant features, and show some feature importance metric. For example, this importance metric comparison is made in "Artificial Intelligence Predictive Model for Hormone Therapy Use in Prostate Cancer".

In conclusion, the paper offers a compelling blend of innovation and potential clinical relevance. To ensure its contribution is both rigorous and meaningful, we recommend the authors to address the outlined concerns.

Reviewer #3 (Remarks to the Author): Expert in immunotherapy biomarkers and cancer genomics

Overall, this is an interesting manuscript that adapts several existing machine learning architectures to predict PDL1 expression across a variety of tumors. The novel component of this framework is integration of a teacher-student relearning component. The authors utilize the TCGA cohort of over 9,000 digital histopathologic images with matched mRNA expression. However, there are several notable weaknesses in the study:

1. Using a 60/14/25 split, the ML models reach an average weighted AUC of 0.83 in internal validation. While the TCGA represents a large cohort of patients/samples, the performance of the external validation using data from CPTAC (figure 2b) falls dramatically to AUCs below 0.6. This is a major limitation to the generalizability of the proposed methods and suggests overfitting/overtraining of the data on the TCGA cohort.
2. It is unclear how the authors handled images from the same patient in the split sample, did they ensure that all images from a single patient would enter the same group (either training, validation, or testing)? This need to be clarified and if it was not done in this manner, it could further contribute to overfitting in the internal validation component.
3. The description of the data utilized is overall poor (especially for the external validation cohort - there is no mention of the sample size or other characteristics of the included patients). The authors also use many abbreviations throughout the figures that are not defined within the figure legends, making the manuscript more difficult to follow.
4. No attempt is made to validate or integrate IHC based imaging (focuses on gene expression), but it is unclear if this will perform well and/or map to H&E findings.

Overall interesting manuscript and approach with some limitations.

RE: NCOMMS-23-34865, “Teacher-student collaborated multiple instance learning for pan-cancer PDL1 expression prediction from histopathology slides”

We sincerely appreciate the valuable feedback and suggestions from the editor and the reviewers regarding our work. Based on the reviewers’ constructive comments we conducted a substantial number of additional experiments and analyses and have revised the manuscript accordingly. We hope that the revisions will fully address reviewers’ concerns.

The main changes in the revised manuscript are as followings:

- We have analyzed n=2,966 FFPE slides from 9 cancer types. These analyses demonstrate that the MILTS can also be applied to standard diagnostic FFPE specimens, and that analysis of such slides recovers concordant histopathological patterns.
- We have performed PDL1 IHC validation on n=20 cases with matched H&E staining. These data reveal a high correlation of predicted PDL1 expression and IHC in each case.
- The manuscript and supplementary materials now provide additional details on hyperparameter configurations, data augmentation settings, and outcomes achieved at various thresholds for dichotomization of PDL1 expression.
- In summary, this revision contains 3 new Figures, 4 new Supplementary Figures and 6 new Supplementary Tables, as well as the aforementioned and further minor amendments to the main text in response to specific reviewer queries.

In the following we provide detailed point-by-point replies to all reviewer comments. We use italic fonts for the reviewer comments and roman font for our replies. Changes made to the manuscript are highlighted in **red**.

Referee 1

Jin et al employ a novel extension of multiple instance learning wherein the probabilities of the individual tiles are optimized to predict slide-level labels of PD-L1 expression. Extensive comparisons are made to other deep learning based multiple instance learning algorithms. The study is impressively comprehensive, involving almost 10,000 slides across 20 tumor types. The figures are well done and clearly convey the results of the study. The histological correlations are intriguing and suggest new avenues of research.

However, there are a couple points that should be explored more carefully:

- 1. The study was conducted on fresh-frozen tissue which is not clearly mentioned until later in the manuscript. It is well known that histological morphology in FFPE can differ greatly from fresh-frozen tissue, degrading the histological correlation conclusions of the study. The utility of the algorithm also becomes into question as it is not applied to a routine diagnostic modality. Would it be possible to better characterize how well or how much further training would be required to achieve high performance on FFPE slides?*

Thank you for your valuable comment. It is important to note that FFPE (Formalin-Fixed Paraffin-Embedded) specimens represent the diagnostic standard and are readily available in routine clinical settings. To further validate our histological conclusions and assess the utility of our proposed model, we conducted the experiments on FFPE tissue samples as you suggest.

Firstly, we retrained the proposed model using FFPE slides sourced from TCGA, focusing on tumors with PDL1 as an established biomarker. The tertile threshold was chosen to demonstrate the performance of the model on FFPE slides. We maintained consistent cohorts splits employed for fresh-frozen sections. The results obtained from the analysis of FFPE slides are presented below,

Table S5

QUANTITATIVE RESULTS OF THE PROPOSED MODEL ON FFPE SLIDES AT TERTILE THRESHOLD.

	BLCA	CESC	COAD	KIRP	LUAD	LUSC	SKCM	STAD	TNBC
AUC	0.81	0.80	0.84	0.83	0.71	0.61	0.71	0.82	0.54
Accuracy	0.76	0.75	0.82	0.81	0.67	0.61	0.62	0.79	0.61
Sensitivity	0.78	0.77	0.67	0.67	0.72	0.54	0.82	0.70	0.54
Specificity	0.77	0.73	0.94	0.84	0.63	0.70	0.52	0.80	0.62

Fig. 1c. The plot illustrates the model's performance on FFPE slides and fresh-frozen slides for the aforementioned tumors, separately.

The model was able to achieve an average AUC of 0.74 on FFPE slides, and the trend in tumor-specific performance was consistent with that of fresh-frozen slides. Nonetheless, there remained a performance gap between FFPE slides and fresh-frozen ones as shown in Fig. 1c, with an AUC of 0.83 for fresh-frozen ones. The finding that frozen slides usually yield better molecular inference aligns with observations reported in several previous studies [1], [2] that focus on computational pathology. Furthermore, research in the field of protein analysis [4] and sequencing [5] has also highlighted the advantages of fresh-frozen slides over FFPE in preserving DNA, RNA, and native proteins.

In addition, we employed the model trained on fresh-frozen slides to predict the outcomes of FFPE slide embeddings and the results are shown in the following table,

Table S6
QUANTITATIVE RESULTS ON FFPE SLIDES USING THE MODEL TRAINED WITH FRESH-FROZEN SLIDES.

	BLCA	CESC	COAD	KIRP	LUAD	LUSC	SKCM	STAD	TNBC
AUC	0.68	0.73	0.72	0.31	0.64	0.67	0.68	0.80	0.50
Accuracy	0.63	0.64	0.67	0.59	0.65	0.68	0.65	0.74	0.61
Sensitivity	0.79	0.89	0.80	0.31	0.70	0.63	0.73	0.76	0.38
Specificity	0.52	0.49	0.57	0.71	0.62	0.71	0.64	0.73	0.74

The aforementioned results, along with existing research, collectively confirm the discrepancy between FFPE slides and fresh-frozen ones, particularly concerning molecular content. Further investigations are warranted to explore strategies for bridging the gap between these two modalities.

Furthermore, we identified consistent recurring patterns for PDL1 positive/negative expression in FFPE slides, which mirror those observed in fresh-frozen slides. These patterns include inflammatory areas, cribriform growth patterns, adenomatous regions, normal colonic crypts, and necrosis. Examples are presented in supplementary Figure S9 in the revised manuscript.

Fig. S9. Typical patterns for PDL1 high/low expression in FFPE slides of COAD. a, Example slides with typical PDL1 positive and **b,** negative patterns.

We have incorporated a relevant discussion of the results obtained from FFPE slides into the revised manuscript and supplementary materials as follows:

Manuscript, p.1: “The approach is evaluated on 12,299 slides across 20 types of solid tumors from TCGA and CPTAC. Among 9 tumors for which PDL1 expression serves as an established biomarker, MILTS achieved a weighted average area under curve of 0.83 on fresh-

frozen and 0.74 on formalin fixed paraffin embedded (FFPE) specimens.”

Manuscript, p.6: “Results demonstrate there exists a salient morpho-transcriptomic link across certain cancer types, whose treatment landscape and prognosis are associated with PDL1 expression, with a weighted average AUC of 0.83 on fresh-frozen and 0.74 on FFPE specimens.”

Manuscript, p.9: “Separate models were trained on FFPE samples using the tertile threshold. We maintained consistent cohort splits employed for fresh-frozen sections and an average AUC of 0.74 was achieved with the model trained with FFPE slides, where the trend in tumor-specific performance was consistent with that of fresh-frozen slides. Detailed results are presented in Supplementary Tables S5 and S6. Nonetheless, there remained a performance gap between FFPE slides and fresh frozen ones as shown in Fig. 1c. The finding that frozen slides usually yield better molecular inference aligns with observations reported in several previous studies [26, 32]. Further investigations are warranted to explore strategies for bridging the gap between these two modalities.”

Manuscript, p.13-14: “The aforementioned recurring patterns were identified in both fresh-frozen and FFPE slides. Additional examples can be found in Figures S8 and S9.”

REFERENCES

- [1] Fu, Y. et al. Pan-cancer computational histopathology reveals mutations, tumor composition and prognosis. *Nat. Cancer* **1**, 800–810 (2020).
- [2] Kather, J. N. et al. Pan-cancer image-based detection of clinically actionable genetic alterations. *Nat. Cancer* **1**, 789–799 (2020).
- [3] Bockmayr, T. et al. Multiclass cancer classification in fresh frozen and formalin-fixed paraffin-embedded tissue by DigiWest multiplex protein analysis. *Laboratory Investigation* **100**, 1288-1299 (2020).
- [4] Gao, X. et al. Comparison of fresh frozen tissue with formalin-fixed paraffin-embedded tissue for mutation analysis using a multi-gene panel in patients with colorectal cancer. *Frontiers in Oncology* **10**, 310 (2020).

2. *Was the sequencing performed on the same block of tissue that the digitized slide was made from? Could that confound the observed histological correlations?*

For the fresh-frozen slides, analytes for sequencing and slide image scanning are from the same vial of the tissue. The relevant documentation within the TCGA encyclopedia outlines the barcoding process, indicating that tissue samples from participants are initially divided into vials. Subsequently, these vials are further partitioned into different portions for characterization and sequencing [1]. According to the biospecimen information available for TCGA cases, both the digitized slides and sequencing data are derived from the same vial of the same sample. And in most cases, these data originate from the same portion, with the slides typically extracted from either the top or bottom layer of the corresponding section. As a result, the spatial proximity of the sequencing data to the tissue contained within the slide is maintained, even though the sequencing data is not generated directly from the tissue present on the slide.

In addition, considering the imperfect matching between sequencing data and digitized slides, we implemented a thresholding strategy in the prediction task, which turns numeric labels into categorical labels (positive/negative). This strategy converted the task into a classification problem rather than a regression one, thereby enhancing the alignment between the data and labels and mitigating the introduction of significant noises to the model.

We have updated the Methods section in the updated manuscript as follows:

Manuscript, p.21: “According to the biospecimen information available for TCGA cases, it is confirmed that the digitized fresh-frozen slides and sequencing data usually originate from the same vial of the same sample, with the slides typically obtained from either the top or bottom layer of the corresponding section. However, FFPE slides are sampled from a different vial and the potential mismatch with the sequencing data could be more pronounced. Thus a thresholding strategy was implemented on sequencing data converting the task into a classification task rather than a regression one, thereby enhancing the alignment between the data and labels.”

REFERENCES

- [1] TCGA Barcode.
https://docs.gdc.cancer.gov/Encyclopedia/pages/TCGA_Barcode.

3. *The study predicts PD-L1 expression as characterized by mRNA. While Pare et al suggests a relationship between PD1 mRNA abundance and response to therapy, they do not find a correlation to PD-L1 tumor expression via IHC: "Second, we investigated the correlation of PD1 mRNA with PDL1 IHC in 74 evaluable samples (63%) from the validation dataset. No correlation between the two biomarkers was found ($r = -0.04$)" If their finding is robust and PD1 mRNA abundance is a better biomarker of response to therapy, then the study should be re-worded to emphasize that the predictive task is based on an mRNA PD1 abundance target as it is anyway suggested to be better associated to outcomes than PD-L1 expression by IHC.*

Thank you for your comment. We chose to demonstrate the utility of our proposed model by predicting PDL1 expression because PDL1 expression was one of the first established biomarkers in the anti-PD1 therapy. Moreover, PDL1 IHC assays have received FDA approval for use in various cancer types in the context of anti-PD1/PDL1 therapy, with numerous studies confirming their diagnostic accuracy [1]-[4].

Additionally, as highlighted in the research by Pare et al., mRNA PD1 abundance holds potential as a biomarker for anti-PD1 monotherapy, showing a strong correlation with the objective response rate (ORR) following such treatment. And they claimed that their in-house dataset of 74 samples did not reveal a significant association between PDL1 tumor expression by IHC and mRNA PD1 abundance. It is worth considering that the scale of the dataset may influence these findings, as demonstrated in other research by Lu et al. [1], who achieved an AUC of 0.65 for PDL1 IHC in correlation with anti-PD1/PDL1 response, analyzing a larger dataset of 8135 cases.

Nevertheless, the diagnostic significance of mRNA PD1 abundance is not against that of PDL1 and should not be underestimated. Its potential as a biomarker for anti-PD1 therapy remains an area of keen interest. In our future research endeavors, we will thoroughly investigate the feasibility of predicting mRNA PD1 abundance based on digitized slides.

REFERENCES

- [1] Lu, S. *et al.* Comparison of Biomarker Modalities for Predicting Response to PD-1/PD-L1 Checkpoint Blockade. *JAMA Oncology* **8**, 1195-1204 (2019).
- [2] Doroshov, D. B. *et al.* PD-L1 as a biomarker of response to immune-checkpoint inhibitors. *Nat. Rev. Clin. Oncol.* **18**, 345–362 (2021).

- [3] Reck, M. et al. Pembrolizumab versus chemotherapy for PD-L1-positive non-small-cell lung cancer. *N. Engl. J. Med.* **375**, 1823–1833 (2016).
- [4] Choueiri, T. K. et al. Correlation of PD-L1 tumor expression and treatment outcomes in patients with renal cell carcinoma receiving sunitinib or pazopanib: Results from COMPARZ, a randomized controlled trial. *Clin. Cancer Res.* **21**, 1071–1077 (2015).

4. *Minor points: References should be provided to other PD-L1 IHC prediction studies including Shamai et al (<https://doi.org/10.1038/s41467-022-34275-9>) and Sha et al (https://doi.org/10.4103/jpi.jpi_24_19).*

Thank you for the suggestion. We have now incorporated a discussion of the research conducted by Shamai et al and Sha et al into the Introduction section of the revised manuscript.

Using IHC as spatial ground truth, both abovementioned studies employed ResNet-like CNN models for predicting PDL1 status. Specifically, Shamai et al focused on breast cancer and trained their model using data pairs consisting of H&E tissue microarray (TMA) images and corresponding IHC-stained TMA images for PDL1. Predictions were made at a resolution of 512×512 pixels, and a fully-supervised approach was applied. On the other hand, Sha et al concentrated on whole-slide images and manually annotated 130 non-small cell lung cancer slides based on PDL1 IHC-stained slides. This approach also implied the use of a fully-supervised training strategy.

In contrast to these methods, our work further demonstrates that it is feasible to predict PDL1 status using only bulk mRNA measurements in a weakly-supervised framework. The proposed method does not need an accurate reference for localized PDL1 expression, such as at the scale of a single patch. Instead, only the global expression label for entire slide images, which is readily accessible in clinical settings, is utilized in training the model. Heterogeneity within the slides can also be highlighted in the patch-level inference, facilitating the alignment of morphological patterns with the biomarker. Consequently, in this way the annotation burden could be substantially reduced.

We have updated the Introduction section in the updated manuscript as follows:

Manuscript, p.5: “Some studies have resorted to manual data annotation and adopted fully supervised learning strategies. For example, in the works of Sha et al. [40] and Shamai et al. [33], PDL1 expression quantification was performed at the tile level by pathologists using paired IHC slides providing an accurate reference for training. However, the dataset size could be constrained due to the labor-intensive nature of annotating such data.”

5. *Minor points: Could a different type of plot be used in Fig 1B? I realize there is already an abundance of bar plots in this figure, but a horizontal bar chart may better convey the composition of the dataset.*

Thank you for the suggestion. We have modified the pie chart in Fig 1b to a horizontal bar chart to better demonstrate the dataset composition, which is shown both here and in the revised manuscript.

Fig. 1

6. *Minor points: Are there any other studies that link histological features with PD-L1 expression as a way of further interrogating your findings?*

Thank you for your suggestion. Upon reviewing related research, we observed that the majority of existing studies have primarily concentrated on investigating the relationship between PDL1 expression and various clinicopathological and molecular features. These include MSI status [1], tumor mutation burden [2], and specific oncogenic driver mutations such as TP53 and EGFR, particularly in the context of lung cancers [3].

Furthermore, in our search, we identified certain studies that incorporated discussions about histological features in relation to PDL1 expression. For example, Roberts et al. [4] discovered that in microsatellite instability-high intestinal adenocarcinoma subtypes, PDL1 expression was predominantly observed on tumor-associated inflammatory cells by pairwise comparison with IHC slides. This resonates with our findings wherein PDL1 expression was consistent with mixed inflammatory stroma. Additionally, predictions from our model demonstrated a positive correlation with the presence of M1 macrophages, which are often associated with a mixed inflammatory infiltrate. Besides, Ng et al. [5] reported that pleomorphic features were significantly associated with elevated PDL1 expression in non-small-cell lung carcinomas.

We have added the related discussion about histological features mentioned in the studies in the revised manuscript on p.13 as follows:

“The findings in [51] align with our observations, where PDL1 expression was predominantly observed on tumor-associated inflammatory cells by pairwise comparison with IHC slides in MSI-H subtypes.”

REFERENCES

- [1] Cho, Y. A. et al. PD-L1 expression is significantly associated with tumor mutation burden and microsatellite instability score. *Cancers* **13**, 4659 (2021).
- [2] Huang, R., James H. et al. A pan-cancer analysis of PD-L1 immunohistochemistry and gene amplification, tumor mutation burden and microsatellite instability in 48,782 cases. *Modern Pathology* **34**, 252-263 (2021).
- [3] Lee, S. E. et al. Association with PD-L1 expression and clinicopathological features in 1000 lung cancers: a large single-institution study of surgically resected lung cancers with a high prevalence of EGFR mutation. *International Journal of Molecular Sciences* **20**, 4794 (2019).
- [4] Roberts, J. et al. PD-L1 expression patterns in microsatellite instability-high intestinal adenocarcinoma subtypes. *American Journal of Clinical Pathology* **152**, 384-391 (2019).
- [5] Ng, K. K. et al. Expression of PD-L1 correlates with pleomorphic morphology and histological patterns of non-small-cell lung carcinomas. *Histopathology* **7**, 1024-1032 (2018).

7. *Minor points: What was the purpose of characterizing PD-L1 expression in tumors where immunotherapy is not heavily used? Why was the usage of immunotherapy in these tumors correlated to the performance of the algorithm?*

Thank you for the comment. Our decision to conduct experiments on tumors where immunotherapy is not heavily used was driven by the objective of conducting an exploratory investigation. We aimed to assess whether there would be differences in prediction performance when comparing these tumors to tumors in which PDL1 serves as an established biomarker.

In general, the tumor types where immunotherapy is infrequently employed tend to exhibit low levels of PDL1 expression. Hence the residual expression can be only weakly predicted from H&E due to the limited PDL1 signal. The low levels of PDL1 expression also suggest that PDL1 inhibitors may have limited therapeutic efficacy in these tumor types. Consequently, the efficacy of immunotherapy was correlated with the morphological prediction performance of the algorithms.

8. *Minor points: L64: It is inferred that IHC is comparably expensive to RT-qPCR which is not true.*

Thank you for the comment. The statement in Line 64 may have caused ambiguity. Our intention with this sentence was to emphasize that both IHC and RT-qPCR are generally more costly in comparison to H&E slides. We have modified the statement in manuscript p.4 as follows:

“Besides, quantification of mRNA expression using techniques like real-time reverse transcription polymerase chain reaction (RT-qPCR) and IHC tests can be costly and time-consuming.”

9. *Minor points: What is the threshold for dichotomization?*

We applied varying thresholds to the tumors based on their mRNA distributions. Specifically, we considered three quantiles for each cancer type: quartile, tertile, and median points. The specific FPKM-UQ values are shown in the table below,

Table S1
QUANTILE VALUES FOR PDL1-RELEVANT TUMORS

Cancer type	BLCA	CESC	COAD	KIRP	LUAD	LUSC	SKCM	STAD	TNBC
Quartile	3.0	6.1	1.4	2.2	4.7	7.2	2.3	2.2	2.3
Tertile	2.0	4.2	1.1	1.6	3.8	5.0	1.7	1.8	2.0
Median	1.0	2.4	0.7	1.2	2.5	2.9	1.0	1.2	1.4

Table S2
QUANTILE VALUES FOR OTHER TUMORS

Cancer type	THCA	READ	HNSC	TGCT	MESO	PRAD	ESCA	UCEC	OV	ACC	LIHC
Quartile	3.6	1.1	5.0	2.6	2.4	0.8	2.6	0.8	1.1	0.5	0.6
Tertile	3.0	0.9	4.0	2.1	1.6	0.7	2.0	0.7	0.9	0.4	0.4
Median	2.2	0.6	2.4	1.3	1.0	0.6	1.4	0.6	0.6	0.3	0.3

The reason why we applied distinct thresholds across tumors lies in that different tumors originate from unique tissues, leading to potential variations in their gene expression backgrounds. For instance, some tumors might inherently have a higher baseline expression of PDL1 compared to others. Therefore, a specific threshold might be necessary for each tumor type to accurately determine whether PDL1 expression is elevated.

We have added the related explanation about threshold settings in the revised manuscript on p.7 as follows:

“In the context of MILTS, a slide-level label initialized from dichotomised mRNA expression levels is employed to supervise the representation learning for the histopathological image, where three quantiles (quartile, tertile, and median points) for each cancer type were considered. Specific values could be found in Supplementary Tables S1 and S2.”

10. Minor points: Why was the ablation study limited to COAD, SKCM and STAD?

Thank you for the comment. We limited the ablation study to these specific cancer types for computational reasons. These particular cancers exhibited the highest predictive performance and were therefore deemed to be the most informative choices for our ablation study.

11. Minor points: The distinction between M1/M2 macrophages is not well defined in humans and the focus could instead be on the function rather than class.

We recognize and concur with the reviewer's observation regarding the ambiguous distinction between M1 and M2 macrophages in humans. The M1 and M2 classification represents two extremes of a spectrum of potential macrophage activation states. Traditionally, M1 macrophages are understood to be pro-inflammatory, instigating and perpetuating inflammatory responses, while M2 macrophages tend to be anti-inflammatory, contributing to the resolution of inflammation and facilitation of tissue repair. The difficulty in clearly distinguishing between M1 and M2 macrophage could arise from their plasticity, uncertainty of markers, intermediate states, and tissue diversity, which is a complex one in immunology and inflammation research. In our study, this part of analysis was conducted based on the output of CIBERSORT and we did not mean to change the annotation.

Referee 2

The paper offers a novel and effective approach to deep learning on WSIs for bio-marker classification tasks in which bio-marker expression has a significant intra-slide variance. In clinical practice pathologists make diagnosis decisions only after finding many instances of bio-marker expression in the slide. No individual instance can indicate the general slide's bio-marker presence. The paper captures the diagnosis' cumulative and smooth nature by redefining the MIL objective as a local maximum-minimum selection criterion. The criterion is based on a global slide label, as well as the model's current confidence in the local label. Using the student-teacher paradigm to create local "pseudo-labels" is an interesting way to upweight the most impactful patches in each slide. Utilizing the machine-learning innovations for patch-level representation and a transformer encoder for slide-level aggregation yields non-trivial results for mRNA-based PDL1 classification in various H&E datasets.

The paper contains clinical findings that reveal links between the slides' morphological features and PDL1. The model performed much better in tumor types where PDL1 is known to have prognostic significance compared to tumor types where its prognostic significance is yet to be proven. Interestingly, upon closer examination, for some cancer types in the group with no proven prognostic significance, the model seems to perform correlatively to the cancers for which there is evidence of prognostic significance. This inter- and intra-group difference implies that tumors that are clinically sensitive to PDL1 blockers have a morphology that is sensitive to the presence of PDL1 and that, in some specific tumor types (HNSC, MESO, TGCT), there is a morphological sensitivity to PDL1 that might indicate as-of-yet unfound prognostic significance.

Another interesting finding is the link between the models' localized labels and known morphological features that characterize PDL1-positive slides in colon cancer. This link increases the credibility of the link between mRNA estimation of PDL1 and pathological diagnosis based on morphological features.

The paper isn't free of weaknesses. The models ground truth for training and performance evaluation uses mRNA for predicting PDL1 expression. Use of mRNA in such a manner is not part of practical clinical protocols. In fact, IHC staining is the current method used for determining the biomarker's presence. Thus, immediate clinical implications are in question.

0. An important issue is the seemingly liberal changes in mRNA dichotomization during test

time. It seems that at some point during the research, the model was tested on three different label quantization levels according to different quantiles of mRNA expression. Only after observing optimal results for the model was the tertile chosen as the dichotomization point. If this is true, it means there was substantial data leakage into the test data before any further experiments were conducted.

Thank you for your valuable comment. To clarify, we did not choose a threshold based on test-time results. The reason we employed multiple thresholds was to investigate and compare the morpho-transcriptomic correlation under varying mRNA levels. These thresholds were consistently applied to both the training and testing stages for all tumors (Tables S1 and S2), and the corresponding results have been presented in the manuscript and supplementary materials (Tables S3, S4, S8-S10).

Table S1
QUANTILE VALUES FOR PDL1-RELEVANT TUMORS

Cancer type	BLCA	CESC	COAD	KIRP	LUAD	LUSC	SKCM	STAD	TNBC
Quartile	3.0	6.1	1.4	2.2	4.7	7.2	2.3	2.2	2.3
Tertile	2.0	4.2	1.1	1.6	3.8	5.0	1.7	1.8	2.0
Median	1.0	2.4	0.7	1.2	2.5	2.9	1.0	1.2	1.4

Table S2
QUANTILE VALUES FOR OTHER TUMORS

Cancer type	THCA	READ	HNSC	TGCT	MESO	PRAD	ESCA	UCEC	OV	ACC	LIHC
Quartile	3.6	1.1	5.0	2.6	2.4	0.8	2.6	0.8	1.1	0.5	0.6
Tertile	3.0	0.9	4.0	2.1	1.6	0.7	2.0	0.7	0.9	0.4	0.4
Median	2.2	0.6	2.4	1.3	1.0	0.6	1.4	0.6	0.6	0.3	0.3

Table S3

QUANTITATIVE RESULTS OF THE PROPOSED MODEL ON PDL1 CLINICALLY RELEVANT TUMORS AT MEDIAN THRESHOLD.

	LUAD	LUSC	BLCA	TNBC	CESC	STAD	COAD	KIRP	SKCM
AUC	0.656 (0.631~ 0.680)	0.650 (0.619~ 0.680)	0.773 (0.740~ 0.802)	0.624 (0.580~ 0.669)	0.841 (0.815~ 0.866)	0.751 (0.724~ 0.776)	0.854 (0.831~ 0.876)	0.605 (0.563~ 0.647)	0.872 (0.853~ 0.891)
Accuracy	0.648 (0.626~ 0.669)	0.633 (0.606~ 0.656)	0.726 (0.696~ 0.756)	0.617 (0.578~ 0.660)	0.737 (0.704~ 0.773)	0.701 (0.676~ 0.723)	0.789 (0.766~ 0.811)	0.603 (0.566~ 0.641)	0.718 (0.689~ 0.821)
Sensitivity	0.585 (0.503~ 0.625)	0.532 (0.359~ 0.644)	0.599 (0.518~ 0.650)	0.523 (0.348~ 0.627)	0.957 (0.886~ 0.980)	0.487 (0.445~ 0.676)	0.753 (0.671~ 0.894)	0.687 (0.348~ 0.801)	0.945 (0.688~ 0.967)
Specificity	0.716 (0.676~ 0.795)	0.736 (0.622~ 0.893)	0.844 (0.803~ 0.910)	0.738 (0.639~ 0.895)	0.628 (0.583~ 0.707)	0.918 (0.720~ 0.943)	0.829 (0.674~ 0.900)	0.521 (0.373~ 0.835)	0.623 (0.582~ 0.872)

Data enclosed in the parentheses represent the corresponding lower bound and the upper bound of 95% confidence interval. The intervals were obtained by bootstrapping strategy with 2000 random resamples.

Table S4

QUANTITATIVE RESULTS OF THE PROPOSED MODEL ON PDL1 CLINICALLY RELEVANT TUMORS AT THE THRESHOLD OF UPPER QUARTILE.

	LUAD	LUSC	BLCA	TNBC	CESC	STAD	COAD	KIRP	SKCM
AUC	0.725 (0.700~ 0.750)	0.690 (0.657~ 0.720)	0.868 (0.841~ 0.892)	0.586 (0.540~ 0.631)	0.899 (0.878~ 0.918)	0.870 (0.848~ 0.890)	0.862 (0.839~ 0.883)	0.809 (0.764~ 0.851)	0.817 (0.785~ 0.850)
Accuracy	0.689 (0.668~ 0.739)	0.644 (0.597~ 0.722)	0.754 (0.720~ 0.787)	0.586 (0.476~ 0.625)	0.824 (0.794~ 0.852)	0.744 (0.721~ 0.769)	0.802 (0.767~ 0.85)	0.750 (0.718~ 0.782)	0.723 (0.697~ 0.845)
Sensitivity	0.762 (0.647~ 0.800)	0.741 (0.575~ 0.811)	0.898 (0.845~ 0.936)	0.557 (0.472~ 0.781)	0.911 (0.850~ 0.942)	0.874 (0.831~ 0.914)	0.743 (0.592~ 0.809)	0.801 (0.739~ 0.864)	0.806 (0.582~ 0.854)
Specificity	0.662 (0.636~ 0.767)	0.606 (0.525~ 0.763)	0.704 (0.663~ 0.754)	0.607 (0.343~ 0.674)	0.788 (0.750~ 0.843)	0.716 (0.689~ 0.743)	0.830 (0.769~ 0.960)	0.736 (0.697~ 0.772)	0.700 (0.669~ 0.919)

Data enclosed in the parentheses represent the corresponding lower bound and the upper bound of 95% confidence interval. The intervals were obtained by bootstrapping strategy with 2000 random resamples.

Table S8

QUANTITATIVE RESULTS OF THE PROPOSED MODEL ON OTHER TUMORS AT THE THRESHOLD OF UPPER QUARTILE.

	ACC	ESCA	HNSC	LIHC	MESO	OV	PRAD	READ	TCGT	THCA	UCEC
AUC	0.819	0.496	0.719	0.558	0.707	0.644	0.629	0.828	0.763	0.838	0.641
Accuracy	0.707	0.560	0.630	0.637	0.677	0.584	0.501	0.680	0.811	0.779	0.526
Sensitivity	0.800	0.650	0.790	0.459	0.625	0.664	0.786	0.832	0.693	0.769	0.777
Specificity	0.685	0.514	0.584	0.692	0.697	0.544	0.417	0.651	0.850	0.783	0.439

Table S9

QUANTITATIVE RESULTS OF THE PROPOSED MODEL ON OTHER TUMORS AT THE THRESHOLD OF UPPER TERTILE.

	ACC	ESCA	HNSC	LIHC	MESO	OV	PRAD	READ	TCGT	THCA	UCEC
AUC	0.550	0.621	0.736	0.463	0.673	0.596	0.657	0.818	0.719	0.818	0.614
Accuracy	0.548	0.596	0.682	0.417	0.619	0.524	0.689	0.765	0.736	0.728	0.590
Sensitivity	0.733	0.787	0.830	0.934	0.738	0.881	0.471	0.750	0.632	0.831	0.662
Specificity	0.453	0.488	0.623	0.132	0.542	0.286	0.799	0.769	0.794	0.676	0.555

Table S10

QUANTITATIVE RESULTS OF THE PROPOSED MODEL ON OTHER TUMORS AT THE THRESHOLD OF MEDIAN.

	ACC	ESCA	HNSC	LIHC	MESO	OV	PRAD	READ	TCGT	THCA	UCEC
AUC	0.535	0.646	0.691	0.493	0.672	0.596	0.669	0.804	0.796	0.833	0.638
Accuracy	0.556	0.640	0.658	0.559	0.619	0.612	0.629	0.760	0.736	0.787	0.619
Sensitivity	0.541	0.374	0.545	0.583	0.736	0.696	0.816	0.808	0.563	0.813	0.739
Specificity	0.543	0.937	0.723	0.523	0.542	0.520	0.454	0.741	0.961	0.765	0.505

Specifically, we considered three quantiles for each cancer type: quartile, tertile, and median points, where the specific FPKM-UQ values are shown in the tables above. We implemented PDL1 predictions for all the tumors in parallel (PDL1-relevant group and the unverified group) across all three thresholds. These results were consistent across different thresholds therefore experimental results at the tertile threshold were primarily presented for convenience and avoiding redundancy. The results, derived from different thresholds for various tumors, are detailed in Tables S3, S4, S8-S10 in the supplementary materials.

The original statement regarding the thresholding strategy for mRNA dichotomization may have been ambiguous. We have addressed this concern by providing a more detailed explanation in the revised manuscript as follows:

Manuscript, p.7: “In the context of MILTS, a slide-level label initialized from dichotomised mRNA expression levels is employed to supervise the representation learning for the histopathological image, where three quantiles (quartile, tertile, and median points) for each cancer type were considered. Specific values could be found in Supplementary Tables S1 and S2.”

Manuscript, p.8: “We also evaluated the using other two threshold settings: the upper quartile (top 75%) and median (50%) expression levels in each cancer type. These two alternative thresholds yielded broadly comparable performance with a mean AUC of 0.81 and 0.75, respectively (Supplementary Fig. S3 and Supplementary Tables S3 and S4). In the following, we discuss results at the upper tertile, unless stated otherwise.”

Manuscript, p.16: “The average AUC value is measured at 0.67 and the average accuracy stands at 64%, indicating a moderate level of discriminatory power in distinguishing between different mRNA expression levels. See Supplementary Tables S8-S10 for the details.”

1. To ensure the validity of clinical conclusions, it is necessary to rerun all experiments displayed in Figures 1, 2, and 6 using the mRNA expression upper quantile and median point as the dichotomization point. This rerun is particularly crucial for the nine cancer types that lack proven prognostic significance for PDL1. The current inferior performance of the model on these cancers can be attributed to overfitting the original 11 cancer types caused by a data leak. Therefore, if the tertile was chosen as the dichotomization point after conducting experiments on test data, it is crucial to conduct this rerun.

We appreciate the reviewer's suggestion, and we would like to clarify our approach regarding the threshold selection. Our choice of threshold was not based on the performance of tumors in the PDL1-relevant groups. Instead, for both the 9 tumors with PDL1 as an established biomarker and the 11 tumors lacking proven prognostic significance, we conducted experiments in parallel using all three thresholds (quartile, tertile, and median points) and these thresholds differed among tumors due to their distinctive FPKM-UQ distributions.

The prediction performance of all tumors at these thresholds is documented in Tables S3, 4, 8-10 in the supplementary materials (also listed under the reply of this comment). Upon closer examination of the table, it becomes evident that the classification performance for each tumor remains consistent across different thresholds. This consistency implies that similar conclusions can be drawn using data at the quartile and median points thresholds.

Table S3

QUANTITATIVE RESULTS OF THE PROPOSED MODEL ON PDL1 CLINICALLY RELEVANT TUMORS AT MEDIAN THRESHOLD.

	LUAD	LUSC	BLCA	TNBC	CESC	STAD	COAD	KIRP	SKCM
AUC	0.656 (0.631~ 0.680)	0.650 (0.619~ 0.680)	0.773 (0.740~ 0.802)	0.624 (0.580~ 0.669)	0.841 (0.815~ 0.866)	0.751 (0.724~ 0.776)	0.854 (0.831~ 0.876)	0.605 (0.563~ 0.647)	0.872 (0.853~ 0.891)
Accuracy	0.648 (0.626~ 0.669)	0.633 (0.606~ 0.656)	0.726 (0.696~ 0.756)	0.617 (0.578~ 0.660)	0.737 (0.704~ 0.773)	0.701 (0.676~ 0.723)	0.789 (0.766~ 0.811)	0.603 (0.566~ 0.641)	0.718 (0.689~ 0.821)
Sensitivity	0.585 (0.503~ 0.625)	0.532 (0.359~ 0.644)	0.599 (0.518~ 0.650)	0.523 (0.348~ 0.627)	0.957 (0.886~ 0.980)	0.487 (0.445~ 0.676)	0.753 (0.671~ 0.894)	0.687 (0.348~ 0.801)	0.945 (0.688~ 0.967)
Specificity	0.716 (0.676~ 0.795)	0.736 (0.622~ 0.893)	0.844 (0.803~ 0.910)	0.738 (0.639~ 0.895)	0.628 (0.583~ 0.707)	0.918 (0.720~ 0.943)	0.829 (0.674~ 0.900)	0.521 (0.373~ 0.835)	0.623 (0.582~ 0.872)

Data enclosed in the parentheses represent the corresponding lower bound and the upper bound of 95% confidence interval. The intervals were obtained by bootstrapping strategy with 2000 random resamples.

Table S4

QUANTITATIVE RESULTS OF THE PROPOSED MODEL ON PDL1 CLINICALLY RELEVANT TUMORS AT THE THRESHOLD OF UPPER QUANTILE.

	LUAD	LUSC	BLCA	TNBC	CESC	STAD	COAD	KIRP	SKCM
AUC	0.725 (0.700~ 0.750)	0.690 (0.657~ 0.720)	0.868 (0.841~ 0.892)	0.586 (0.540~ 0.631)	0.899 (0.878~ 0.918)	0.870 (0.848~ 0.890)	0.862 (0.839~ 0.883)	0.809 (0.764~ 0.851)	0.817 (0.785~ 0.850)
Accuracy	0.689 (0.668~ 0.739)	0.644 (0.597~ 0.722)	0.754 (0.720~ 0.787)	0.586 (0.476~ 0.625)	0.824 (0.794~ 0.852)	0.744 (0.721~ 0.769)	0.802 (0.767~ 0.85)	0.750 (0.718~ 0.782)	0.723 (0.697~ 0.845)
Sensitivity	0.762 (0.647~ 0.800)	0.741 (0.575~ 0.811)	0.898 (0.845~ 0.936)	0.557 (0.472~ 0.781)	0.911 (0.850~ 0.942)	0.874 (0.831~ 0.914)	0.743 (0.592~ 0.809)	0.801 (0.739~ 0.864)	0.806 (0.582~ 0.854)
Specificity	0.662 (0.636~ 0.767)	0.606 (0.525~ 0.763)	0.704 (0.663~ 0.754)	0.607 (0.343~ 0.674)	0.788 (0.750~ 0.843)	0.716 (0.689~ 0.743)	0.830 (0.769~ 0.960)	0.736 (0.697~ 0.772)	0.700 (0.669~ 0.919)

Data enclosed in the parentheses represent the corresponding lower bound and the upper bound of 95% confidence interval. The intervals were obtained by bootstrapping strategy with 2000 random resamples.

Table S8

QUANTITATIVE RESULTS OF THE PROPOSED MODEL ON OTHER TUMORS AT THE THRESHOLD OF UPPER QUANTILE.

	ACC	ESCA	HNSC	LIHC	MESO	OV	PRAD	READ	TCGT	THCA	UCEC
AUC	0.819	0.496	0.719	0.558	0.707	0.644	0.629	0.828	0.763	0.838	0.641
Accuracy	0.707	0.560	0.630	0.637	0.677	0.584	0.501	0.680	0.811	0.779	0.526
Sensitivity	0.800	0.650	0.790	0.459	0.625	0.664	0.786	0.832	0.693	0.769	0.777
Specificity	0.685	0.514	0.584	0.692	0.697	0.544	0.417	0.651	0.850	0.783	0.439

Table S9

QUANTITATIVE RESULTS OF THE PROPOSED MODEL ON OTHER TUMORS AT THE THRESHOLD OF UPPER TERTILE.

	ACC	ESCA	HNSC	LIHC	MESO	OV	PRAD	READ	TCGT	THCA	UCEC
AUC	0.550	0.621	0.736	0.463	0.673	0.596	0.657	0.818	0.719	0.818	0.614
Accuracy	0.548	0.596	0.682	0.417	0.619	0.524	0.689	0.765	0.736	0.728	0.590
Sensitivity	0.733	0.787	0.830	0.934	0.738	0.881	0.471	0.750	0.632	0.831	0.662
Specificity	0.453	0.488	0.623	0.132	0.542	0.286	0.799	0.769	0.794	0.676	0.555

Table S10

QUANTITATIVE RESULTS OF THE PROPOSED MODEL ON OTHER TUMORS AT THE THRESHOLD OF MEDIAN.

	ACC	ESCA	HNSC	LIHC	MESO	OV	PRAD	READ	TCGT	THCA	UCEC
AUC	0.535	0.646	0.691	0.493	0.672	0.596	0.669	0.804	0.796	0.833	0.638
Accuracy	0.556	0.640	0.658	0.559	0.619	0.612	0.629	0.760	0.736	0.787	0.619
Sensitivity	0.541	0.374	0.545	0.583	0.736	0.696	0.816	0.808	0.563	0.813	0.739
Specificity	0.543	0.937	0.723	0.523	0.542	0.520	0.454	0.741	0.961	0.765	0.505

Therefore, we chose to primarily present experimental results at the tertile threshold for convenience, as it provides a representative and consistent performance while avoiding redundancy in the presentation of results.

To avoid unnecessary misunderstanding, we have provided a more detailed explanation about the choice of thresholds in the Methods section of the revised manuscript as follows:

Manuscript, p.8: “We also evaluated the using other two threshold settings: the upper quartile (top 75%) and median (50%) expression levels in each cancer type. These two alternative thresholds yielded broadly comparable performance with a mean AUC of 0.81 and 0.75, respectively (Supplementary Fig. S3 and Supplementary Tables S3 and S4). In the following, we discuss results at the upper tertile, unless stated otherwise.”

Manuscript, p.16: “The average AUC value is measured at 0.67 and the average accuracy stands at 64%, indicating a moderate level of discriminatory power in distinguishing between different mRNA expression levels. See Supplementary Tables S8-S10 for the details.”

2. *Threshold-dependent statistics such as accuracy, sensitivity, specificity, and F1 can only be discussed if the reader understands the threshold for model classification. In line 151, Youden's J statistic is mentioned as the method for finding the threshold. Is this statistic methodology used for all experiments? If not, how were the thresholds chosen?*

Thank you for the comment. First, we wish to emphasize that our intention was not to set a predetermined threshold for predicted probabilities in determining positive/negative status. Rather, our objective was solely to benchmark performance across various tumors. So Youden's J statistic was consistently applied in all our experiments to select the optimal output probability threshold, which, in turn, determined the accuracy, sensitivity, and specificity.

To elaborate, Youden's J statistic is a single metric that effectively captures the performance of a dichotomous diagnostic test. It is defined as follows,

$$J = \text{sensitivity} + \text{specificity} - 1 = \frac{TP}{TP+FN} + \frac{TN}{TN+FP} - 1$$

TP , TN , FP , and FN respectively represent counts of true positive, true negative, false positive and false negative predictions. The optimal threshold for output probability is the one that maximizes function J .

We have supplemented related description in the updated manuscript on p.9 as follows:

“To benchmark performance across various tumors, the threshold to determine the accuracy, sensitivity and specificity is selected by Youden’s J statistic. The optimal threshold here is the one that maximizes the sum of sensitivity and specificity. This strategy is applied across all our experiments unless stated otherwise.”

3. *The ablation study mentioned in line 177 should come after some description of the model architecture, as the reader still needs to be made aware of the different modules contained in the model.*

Thank you for the comment. We have now provided additional information about the modules mentioned in the ablation study in the revised manuscript on p.10-11 to ensure a better understanding of our model as below,

“Specifically, MILTS consists of two key modules: a teacher-student MIL module for patch-level feature extraction and a transformer-based feature aggregation module for slide-level predictions. The ablation study was conducted by examining these two modules in isolation. Details about the modules are provided in the Methods section.”

4. *The supplementary material should mention specific parameters used for model augmentation.*

We have supplemented a table documenting parameter setting for data augmentation used in the training of the proposed model as below,

Table S12

PARAMETER SETTINGS FOR DATA AUGMENTATION

Methods	RandomVerticalFlip	RandomRotation	RandomResizedCrop	ColorJitter	Normalize
parameters	DEFAULT	Degree = (-90°, 90)	Size = (224, 224)	brightness=0.2, contrast=0.2, saturation=0.2, hue=0.2	mean=(0.485, 0.456, 0.406) std= (0.229, 0.224, 0.225)

The related description has been added in the revised manuscript on p.22 as follows:

“Detailed parameter settings for data augmentation are shown in Supplementary Table S12.”

5. *Neither the hyperparameters used for training MILTS nor the method by which they are found are mentioned in the paper. The hyperparameters used for comparison with the competing methods (Campella's, CLAM, TransMIL) are included in Table S3, but there is no mention of how these were found. For the comparisons in Figures 2a and 2b to be convincing, the reader needs to be convinced that all models compared had a fair chance.*

Thank you for highlighting the importance of hyperparameter transparency. For the proposed method, the primary hyperparameters were documented in the Implementation details of the Methods section. The related descriptions about patch-level feature extractor are as follows,

“Before the start of MILTS training, we initialized ResNet34 with ImageNet [43] pretrained parameters for both teacher and student model. A mini-batch size of 512 was adopted to accelerate computation and we used a stochastic gradient descent (SGD) optimizer with an initial learning rate of 1×10^{-2} to optimize weights of the models. The strategy of cosine annealing was applied to schedule the learning rate and the minimum value was set as 1×10^{-4} . All the models were trained for 30 epochs, during which two circles of learning rate change were completed.”

Specifically, we adopted the minibatch size of 512 primarily due to our hardware's limitations. More extensive batch sizes could potentially be managed by GPUs with a greater graphic memory capacity. We performed hyperparameter selection using data from COAD, BLCA, and LUSC. The initial learning rate for the patch-level feature extractor was selected from a range of values, including 1×10^{-1} , 1×10^{-2} , 1×10^{-3} and 1×10^{-4} . Additionally, the number of cycles for the learning rate scheduler was chosen from the set [1, 2, 3]. Default settings were retained for other hyperparameters associated with the optimizer and learning rate scheduler.

Furthermore, we have supplemented our manuscript with details regarding other pivotal hyperparameters of our model. Concretely, the proportions of labeled instances within a single slide were defined as 0.25, 0.35, and 0.45 for the quartile, tertile, and median points, respectively. The consistency cost weight λ for the loss function was determined to be 100, following a selection process from the set [0.1, 0.5, 1, 5, 10, 50, 100, 200]. EMA decay α for updating the teacher model was set to 0.99 throughout training.

For the comparison methods, the related hyperparameters are listed in Table S3. Specifically,

for embedding-based MIL methods such as CLAM and TransMIL, the only hyperparameter need to be determined was learning rate. Their optimal learning rates were selected from the set $[1 \times 10^{-2}, 1 \times 10^{-3}, 5 \times 10^{-4}, 2 \times 10^{-4}, 1 \times 10^{-4}, 1 \times 10^{-5}]$. For Campellas’s method, we modified the minibatch size to 512 for the same reason and determined the learning rate from the set $[1 \times 10^{-1}, 1 \times 10^{-2}, 1 \times 10^{-3}, 1 \times 10^{-4}]$.

We have enhanced the Methods section and supplementary materials in our revised manuscript to provide a comprehensive description of hyperparameter settings and selection ranges for both our proposed method and the comparative methods as follows:

Manuscript, p.28: “Before the start of MILTS training, we initialized ResNet34 with ImageNet [44] pretrained parameters for both teacher and student model. A mini-batch size of 512 was adopted to accelerate computation and we used a stochastic gradient descent (SGD) optimizer with an initial learning rate of 1×10^{-2} to optimize weights of the models. The proportions of labeled instances within a single slide were defined as 0.25, 0.35, and 0.45 for the quartile, tertile, and median points, respectively. The consistency cost weight λ for the loss function was determined to be 100. EMA decay α for updating the teacher model was set to 0.99 throughout training. The strategy of cosine annealing was applied to schedule the learning rate and the minimum value was set as 1×10^{-4} . All the models were trained for 30 epochs, during which two circles of learning rate change were completed.”

6. Beginning in line 374, the sentence "This threshold was set based on the median point, upper tertile, and upper quartile of the mRNA quantification distribution" is the only mention of the dichotomization of mRNA used during training time. The paper lacks an exact explanation of how the threshold was chosen in training time.

We appreciate the reviewer's request for a more detailed explanation of the mRNA dichotomization threshold. The upper quartile, upper tertile and median thresholds were determined individually for each tumor type based on the distribution of their mRNA quantification. The threshold was applied consistently to the whole dataset including training set, validation set and test set for each specific tumor type. Specifically, the FPKM-UQ threshold values for each PDL1-relevant tumor are shown in the table below,

Table S1
QUANTILE VALUES FOR PDL1-RELEVANT TUMORS

Cancer type	BLCA	CESC	COAD	KIRP	LUAD	LUSC	SKCM	STAD	TNBC
Quartile	3.0	6.1	1.4	2.2	4.7	7.2	2.3	2.2	2.3
Tertile	2.0	4.2	1.1	1.6	3.8	5.0	1.7	1.8	2.0
Median	1.0	2.4	0.7	1.2	2.5	2.9	1.0	1.2	1.4

Table S2
QUANTILE VALUES FOR OTHER TUMORS

Cancer type	THCA	READ	HNSC	TGCT	MESO	PRAD	ESCA	UCEC	OV	ACC	LIHC
Quartile	3.6	1.1	5.0	2.6	2.4	0.8	2.6	0.8	1.1	0.5	0.6
Tertile	3.0	0.9	4.0	2.1	1.6	0.7	2.0	0.7	0.9	0.4	0.4
Median	2.2	0.6	2.4	1.3	1.0	0.6	1.4	0.6	0.6	0.3	0.3

The reason why we applied distinct thresholds across tumors lies in that different tumors originate from unique tissues, leading to potential variations in their gene expression backgrounds. For instance, some tumors might inherently have a higher baseline expression of PDL1 compared to others. Therefore, a specific threshold might be necessary for each tumor type to accurately determine whether PDL1 expression is elevated.

We have provided a more detailed explanation in the revised manuscript as follows:

Manuscript, p.7: "In the context of MILTS, a slide-level label initialized from dichotomised mRNA expression levels is employed to supervise the representation learning for the histopathological image, where three quantiles (quartile, tertile, and median points) for each cancer type were considered. Specific values could be found in Supplementary Tables S1 and S2."

7. *The sentence beginning in line 356, "A commonly used split for data, in which 60% is reserved for training, 15% for validation, and 25% for testing, was employed for the majority of the tumors analyzed." raises questions as to the splitting strategy. The paper needs to mention whether slides taken from patients included in the training and validation sets were also found in the test set. If so, the split may cause data leakage from the test set and raise concerns about unmeaningful results. One quick and imperfect way to remedy this split problem is to remove patients in the test set that also appear in the training set. If such a data contamination exists, there is a need to redo the split properly and re-run all experiments.*

We appreciate the reviewer's diligence in scrutinizing our data splitting strategy. It is indeed crucial to ensure the integrity of the dataset to prevent any potential data leakage. For all the experiments described in this paper, the datasets were meticulously partitioned based on patient IDs, ensuring that there was no contamination. This approach was employed to ensure the complete separation of patients between the training, validation, and test sets, thus avoiding potential risks of data leakage.

We have supplemented related descriptions about data split in the Methods section as follows:
“A commonly used split for data, in which 60% is reserved for training, 15% for validation, and 25% for testing, was employed for the majority of the tumors analyzed. It’s worth noting that the splitting was performed based on patient IDs. This led to a distribution of 5,373 slides for training, 1,189 slides for validation, and 3,182 slides for testing in the case of fresh-frozen slides, and 1,837, 386, and 743 slides respectively for FFPE slides.”

8. The claims made using Figure 5b are unconvincing. A simple rank correlation between 2 features does not support the claim that the features added unique data previously unavailable. One way to show this claim is accurate is to try to fit a model on the slide representation token, concatenated with the clinically significant features, and show some feature importance metric. For example, this importance metric comparison is made in "Artificial Intelligence Predictive Model for Hormone Therapy Use in Prostate Cancer".

Thank you for the valuable suggestion. To further verify the correlation, we crafted three distinct feature sets, consisting of (i) the deep features derived from the proposed model, (ii) deep features concatenated with standardized clinical features and (iii) clinical features alone. For this analysis, we employed an XGBoost classifier to conduct the prediction based on the abovementioned embeddings, using the COAD data as a representative example. The classification results are presented below,

Fig. 6c. Classification performance by the deep features, deep features concatenated with standardized clinical features and clinical features alone.

From the results, it is evident that the deep features extracted by MILTS outperform the purely clinical features as depicted in Fig. 6b of the manuscript. Additionally, these results highlight that the deep features extracted by our model are distinct and do not overlap with or duplicate the information of clinical features. Instead, they encapsulate unique morphological information sourced directly from the pathological slides.

In addition, we also implemented SHAP, which was used in "Artificial Intelligence Predictive Model for Hormone Therapy Use in Prostate Cancer", to showcase the relative importance of the deep features in comparison to clinical features. Using SHAP values, it could be observed that the deep features derived from the proposed model hold significantly greater importance

than the clinical features (ones with indices greater than 523). This not only underscores the superior predictive capacity of the deep features for expression but also highlights their low correlation with clinical attributes.

Fig. 6d. SHAP Feature Importance Analysis on MILTS features and clinical features. Features with indices greater than 523 correspond to clinical features, while those with indices less than or equal to 523 represent deep features extracted by MILTS.

The related analysis and figures have been added to Results section in the revised manuscript as follows:

“To further verify the correlation, we crafted three distinct feature sets, consisting of (i) the deep features derived from the proposed model, (ii) deep features concatenated with standardized clinical features and (iii) clinical features alone. An XGBoost classifier was employed to conduct the prediction based on the forementioned embeddings. The results are presented in Fig. 6c. It’s evident that the deep features extracted by MILTS outperform the purely clinical features which implies that the proposed model encapsulates unique morphological information sourced directly from the pathological slides. We also implemented SHAP to showcase the relative importance of the deep features in comparison to clinical features (Fig 6d). Using SHAP values, it could be observed that the deep features derived from the proposed model hold significantly greater importance than the clinical features.”

Referee 3

Overall, this is an interesting manuscript that adapts several existing machine learning architectures to predict PDL1 expression across a variety of tumors. The novel component of this framework is integration of a teacher-student relearning component. The authors utilize the TCGA cohort of over 9,000 digital histopathologic images with matched mRNA expression. However, there are several notable weaknesses in the study:

- 1. Using a 60/15/25 split, the ML models reach an average weighted AUC of 0.83 in internal validation. While the TCGA represents a large cohort of patients/samples, the performance of the external validation using data from CPTAC (figure 2b) falls dramatically to AUCs below 0.6. This is a major limitation to the generalizability of the proposed methods and suggests overfitting/overtraining of the data on the TCGA cohort.*

Thank you for the valuable comments. We appreciate the reviewer's concern regarding the external validation using CPTAC data, which is indeed a critical aspect of our study. We discussed with this discrepancy between internal and external validation results in the Discussion section of our manuscript.

It is worth noting that the challenges related to generalization in the field of computational pathology have been widely recognized and discussed. Variations in scanners, staining protocols, image compression artifacts, and other factors often introduce biases that can affect the performance of deep learning models [1]. Usually, in the context of tasks that pathologists routinely perform, such as tumor-versus-normal classification and lung cancer subtyping, deep learning models tend to exhibit better generalization in external cohorts. This improved generalization can be attributed to the presence of distinctive morphological features that different tissues exhibit, enabling models to discern and adapt more effectively. However, when it comes to molecular-level tasks, like gene mutation and expression prediction, the complexity arises from the scarcity of explicit patterns. In such cases, models often encounter challenges in generalizing as expected [3]. Similar observations have been reported in other studies within the field of computational pathology [4]-[8], underscoring the shared recognition of these challenges and the need for further research to address them effectively.

In our cases, apart from the potential factors previously mentioned that could influence generalization, we would like to highlight that the CPTAC COAD and BRCA dataset used in our validation consists of a combination of fresh-frozen and FFPE slides. The challenge of

feature transferring between these two modalities is a well-recognized concern within the field of computational pathology. In addition, during our investigation into mRNA distribution within the TCGA and CPTAC datasets, we also identified discrepant patterns as shown in Fig. S12, which may arise from differences in quantification protocols or variations in patient populations. This discrepancy may imply a potential mismatch between bulk RNA sequences and the tissue in the histopathology slide. While the proposed model still demonstrated a modest advantage over the comparison methods in external validation, it is important to acknowledge that the observed gap in performance could be attributed to the aforementioned issues.

Fig. S12. mRNA distribution of CPTAC and TCGA COAD dataset.

Furthermore, considering the challenges prevalent in this field, we are committed to addressing the generalization issues in our forthcoming research endeavors and we would like to extend our gratitude again to the reviewer for their constructive comments.

REFERENCES

- [1] Van der Laak, J., Litjens, G., & Ciompi, F. Deep learning in histopathology: the path to the clinic. *Nature Medicine* **27**, 775-784 (2021).
- [2] Wang, J. M., et al. Deep learning integrates histopathology and proteogenomics at a pan-cancer level. *Cell Reports Medicine* **4** (2023).
- [3] Chen, M., et al. Classification and mutation prediction based on histopathology H&E images in liver cancer using deep learning. *NPJ Precision Oncology* **4**, 14 (2020).
- [4] Ehle, A., et al. Deep learning in cancer pathology: a new generation of clinical biomarkers. *British Journal of Cancer* **124**, 686-696 (2021).
- [5] Howard, F. M., et al. The impact of site-specific digital histology signatures on deep learning model accuracy and bias. *Nature Communications* **12**, 4423 (2021).
- [6] Fu, Y. et al. Pan-cancer computational histopathology reveals mutations, tumor composition and prognosis. *Nature Cancer* **1**, 800–810 (2020).
- [7] Jeong, Y., et al. Deep learning model to predict Epstein–Barr virus associated gastric cancer in histology.

Scientific Reports **12**, 18466 (2022).

- [8] Kiehl, L., et al. Deep learning can predict lymph node status directly from histology in colorectal cancer. *European Journal of Cancer* **157**, 464-473 (2021).

- 2. It is unclear how the authors handled images from the same patient in the split sample, did they ensure that all images from a single patient would enter the same group (either training, validation, or testing)? This needs to be clarified and if it was not done in this manner, it could further contribute to overfitting in the internal validation component.*

We appreciate the reviewer's diligence in scrutinizing our data splitting strategy. It is indeed crucial to ensure the integrity of the dataset to prevent any potential data leakage. For all the experiments described in this paper, the datasets were meticulously partitioned based on patient IDs. This approach was employed to ensure the complete separation of patients between the training, validation, and test sets, thus avoiding any potential risk of data leakage.

We have supplemented related descriptions about data split in the Methods section as follows: “A commonly used split for data, in which 60% is reserved for training, 15% for validation, and 25% for testing, was employed for the majority of the tumors analyzed. It’s worth noting that the splitting was performed based on patient IDs. This led to a distribution of 5,373 slides for training, 1,189 slides for validation, and 3,182 slides for testing in the case of fresh-frozen slides, and 1,837, 386, and 743 slides respectively for FFPE slides.”

3. *The description of the data utilized is overall poor (especially for the external validation cohort - there is no mention of the sample size or other characteristics of the included patients). The authors also use many abbreviations throughout the figures that are not defined within the figure legends, making the manuscript more difficult to follow.*

Thank you for your comment. We apologize for the inadequate description of the data, especially concerning the external validation cohort. In our revised manuscript, we have taken steps to provide a comprehensive overview of the dataset used in this research. And the explanations for the abbreviations used in the figures have been also provided.

Specifically, we conducted evaluations based on CPTAC COAD and BRCA datasets, both of which include paired mRNA quantification files and slide images. The COAD dataset consists of 220 samples from 100 cases, while the BRCA dataset comprises 394 samples from 117 cases. Characteristics of patients included in the dataset is presented in Table S11. More detailed descriptions are also added to the Methods section in the revised manuscript as follows,

Table S11
PATIENT CHARACTERISTICS OF CPTAC COHORT

Dataset	CPATAC-COAD (n=100)	CPTAC-BRCA (n=117)
Age (year)		
Range	/	31.3-91.3
Average	/	60.7
Sex (n, %)		
Female	58 (58)	106 (90.6)
Male	42 (42)	/
Not reported	/	11 (9.4)
Tumor Stage (n, %)		
Stage I	10 (10)	4 (3.4)
Stage II	41 (41)	70 (59.8)
Stage III	42 (42)	32 (27.4)
Stage IV	7 (7)	/
Not performed/reported	/	11 (9.4)

“Whole slide images and transcriptome profiling data utilized in the experiments comes from TCGA project via National Cancer Institute (NCI) Genomic Data Commons Portal and CPTAC project via the Cancer Imaging Archive (TCIA) Pathology Portal, which comprises 12,299 H&E stained whole slide images and corresponding mRNA quantification data

obtained from 6,715 patients diagnosed across 20 different types of cancer. Data from CPTAC dataset is employed for external validation purposes. The sample types comprise mainly primary solid tumor as well as metastatic tissues. Specifically, the dataset comprises 8,719 fresh-frozen slides and 2,966 FFPE slides from TCGA, along with 614 slides from CPTAC COAD and BRCA. Characteristics of patients is presented in Supplementary Table S11. We collect all available whole slide images scanned at a magnification of $20\times$ or higher, along with their CD274 mRNA expression level read counts normalized by the upper quartile fragments per kilobase of transcript per million mapped reads (FPKM-UQ).

According to the biospecimen information available for TCGA cases, it is confirmed that the digitized fresh-frozen slides and sequencing data usually originate from the same vial of the same sample, with the slides typically obtained from either the top or bottom layer of the corresponding section. However, FFPE slides are sampled from a different vial and the potential mismatch with the sequencing data could be more pronounced. Thus a thresholding strategy was implemented on sequencing data converted the task into a classification problem rather than a regression one, thereby enhancing the alignment between the data and labels. A commonly used split for data, in which 60% is reserved for training, 15% for validation, and 25% for testing, was employed for the majority of the tumors analyzed. It's worth noting that the splitting was performed based on patient IDs. This led to a distribution of 5,373 slides for training, 1,189 slides for validation, and 3,182 slides for testing in the case of fresh-frozen slides, and 1,837, 386, and 743 slides respectively for FFPE slides.”

Fig. 1. The framework of MILTS and its performance on clinically PDL1-relevant tumors. **a**, The training and inference workflow of MILTS. The training mainly includes three steps. First the data of patient cohorts are divided into training set, validation set and test set, which would be followed by patching and random augmentations. Then these tiles are utilized to train the patch-level teacher-student collaborated network in a multiple instance learning manner. At last, the trained patch-level teacher model (or student model) works as the extractor of both statistical features and deep features. The deep features of patches in the same slide are further fused into a slide token and combined with the statistical summary of patch-level features to train an MLP classifier which would give the patient-level diagnosis. **MIL**, multiple instance learning; **S**, student; **T**, teacher; **C**, concatenation; **MLP**, multi-layer perceptron. **b**, Quantities of slide images of different tumors. **c**, The plot illustrating the model's performance on FFPE slides and fresh-frozen slides for the aforementioned tumors, separately.

Furthermore, we have provided explanations for the abbreviations used in the figures within their respective captions. For example, the description for Figure 1 has been added as follows,

And the description for Fig. 7 has been added as follows,

Fig. 7. Performance on other cancer entities. **a**, The histogram of model performance on the other 11 cancer entities indicated by AUC, accuracy, sensitivity and specificity. The average performance of the group with PDL1 diagnostics is displayed in the leftmost column. **w/PDL1** means the tumors with PDL1 as an established biomarker. **b**, The scatter plot showing the correlation between the overall PDL1 expression level and AUC performance. Box plots displayed alongside the Y and X axes, represent the distribution of AUCs and the PDL1 FPKM, respectively. Groups with and without PDL1 diagnosis are distinguished by dark blue and light blue markers in the data. **w/**, with; **w/o**, without.

4. *No attempt is made to validate or integrate IHC based imaging (focuses on gene expression), but it is unclear if this will perform well and/or map to H&E findings.*

We appreciate the reviewer's inquiry regarding the validation on IHC slides. It is indeed crucial to investigate whether the patterns observed in the PDL1 heatmaps generated from H&E slide images align with those from the corresponding IHC slides. To address this, we conducted correlation analysis between model predictions and IHC scores using a set of 20 colon adenocarcinoma samples that we obtained from our collaborators at Charite, Berlin. In this case, we had access to paired H&E slides and IHC slides for comprehensive analysis.

Fig. S1. The Original IHC file, hematoxylin channel, DAB channel and the pseudo fluorescence image generated by the two abovementioned channels.

In particular, we employed a color deconvolution technique [1] to separate the immunohistochemical staining from the hematoxylin counterstaining, allowing us to quantify PDL1 expression in the IHC slide images as shown in Fig. S1. The diaminobenzidine (DAB) channel was extracted from the original IHC tiles, and its pixel-wise intensity was utilized for the quantification of PDL1 expression. Subsequently, we performed image registration between the H&E and IHC slide pairs using SIFT [2] as the feature descriptor to mitigate any shifts and distortions (Fig. S2).

With the registered image pairs in place, we proceeded with regional-level correlation analysis. The entire slide images were partitioned into 100 regions, and we calculated Pearson's correlation coefficient by accumulating the predicted positive probability and diaminobenzidine intensity within each region. The predictions by the model trained at the tertile threshold with COAD FFPE data were adopted here. An example is shown in Fig. 5a to provide a visual comparison between the H&E-based heatmap and the corresponding IHC slide images. Visually, PDL1 IHC levels exhibited similar patterns PDL1 as H&E based predictions. The slide-wise correlation coefficients and corresponding scatter plots illustrating the relationship between IHC scores and predicted positive probability are presented in Fig. 5b. It is worth noting that the model predictions consistently exhibit a strong positive correlation

with IHC quantification, with an average Pearson's correlation coefficient of 0.74 among the 20 samples. Together these findings confirm the model's ability to deconvolve gene expression signals and attribute these signals to distinct histopathological areas.

Fig. S2. Example of registration between the H&E slide image and the corresponding IHC slide image.

Fig. 5a. Visual comparison between predicted heatmaps and corresponding IHC slide images. The stained-separated images are produced by employing the diaminobenzidine and hematoxylin channels from IHC slide images as the green and blue components, respectively. A more pronounced green area signifies higher PDL1 expression.

Fig. 5b. Scatter plots illustrating the relationship between normalized IHC quantification and predicted positive probability by the proposed model.

We have supplemented related discussions about validation on IHC slides in the Results section in the revised manuscript and supplementary materials as follows:

Manuscript, p.14: “Further, the predicted patterns observed of PDL1 were also validated using paired IHC slides. Correlation analysis between model predictions and IHC scores was conducted using a set of 20 colon adenocarcinoma samples. Visually, PDL1 IHC levels exhibited similar patterns PDL1 as H&E based predictions (Fig. 5a). IHC levels were quantified in patches of $128 \times 128 \mu\text{m}^2$ and compared to the H&E based PDL1 prediction in matching areas containing 1% of the total patches on the slide. Model predictions exhibit a consistent strong positive correlation with IHC across all 20 slides, with an average Pearson’s correlation coefficient of 0.74 (Fig. 5b). Together these findings confirm the model’s ability to deconvolve gene expression signals and attribute these signals to distinct histopathological areas. More details are provided in the supplementary material.”

Supplementary Materials, p.2: “**Correlation analysis between model predictions and paired IHC slides.** To conduct the validation on IHC imaging, we collected 20 colon adenocarcinoma samples with paired H&E slides and IHC slides. In particular, we employed a color deconvolution technique [1] to separate the immunohistochemical staining from the

hematoxylin counterstaining, allowing us to quantify PDL1 expression in the IHC slide images as shown in Fig. S1. The diaminobenzidine (DAB) channel was extracted from the original IHC tiles, and its pixel-wise intensity was utilized for the quantification of PDL1 expression. Subsequently, we performed image registration between the H&E and IHC slide pairs using SIFT [2] as the feature descriptor to mitigate any shifts and distortions (Fig. S2).

With the registered image pairs in place, we proceeded with regional-level correlation analysis. The entire slide images were partitioned into 100 regions, and Pearson's correlation coefficient was calculated by accumulating the predicted positive probability and diaminobenzidine intensity within each region. The predictions by the model trained at the tertile threshold with COAD FFPE data were adopted here.”

REFERENCES

- [1] Ruifrok, A. C., & Johnston, D. A. Quantification of histochemical staining by color deconvolution. *Analytical and Quantitative Cytology and Histology* **23**, 291-299 (2001).
- [2] Lowe, D. G. Distinctive image features from scale-invariant keypoints. *International Journal of Computer Vision* **60**, 91-110 (2004).

REVIEWERS' COMMENTS

Reviewer #1 (Remarks to the Author):

The authors' efforts to address the comments are greatly appreciated and the revision is a vastly improved manuscript. Our remaining comment is minor:

The sentence "Currently, PDL1 expression is predominantly quantified by immunohistochemistry (IHC) assays, and some recent research indicates that mRNA expression level also shares a strong correlation with IHC quantification[13]." should be modified to reflect that mRNA expression level was found to be significantly associated with response, not that mRNA expression is correlated to IHC quantification, which better reflects the findings in [13].

Reviewer #3 (Remarks to the Author):

The authors have submitted a revised manuscript with multiple clarifications and including FFPE (in addition to fresh frozen) sample analysis. I find the authors responsive to reviewer feedback and the current manuscript far stronger (and more clear) than initial submission. My major concerns have been addressed.

RE: NCOMMS-23-34865A, “Teacher-student collaborated multiple instance learning for pan-cancer PDL1 expression prediction from histopathology slides”

We sincerely appreciate the valuable feedback and suggestions from the editor and the reviewers regarding our work. We have modified the related description according to the suggestion by review 1 and changes made to the manuscript are highlighted in red. In the following we use italic fonts for the reviewer comments and roman font for our replies.

Referee 1

The authors' efforts to address the comments are greatly appreciated and the revision is a vastly improved manuscript. Our remaining comment is minor:

The sentence "Currently, PDL1 expression is predominantly quantified by immunohistochemistry (IHC) assays, and some recent research indicates that mRNA expression level also shares a strong correlation with IHC quantification[13]." should be modified to reflect that mRNA expression level was found to be significantly associated with response, not that mRNA expression is correlated to IHC quantification, which better reflects the findings in [13].

Thank you for your comment. We have modified the corresponding description in the manuscript p.3 as follows,

"Currently, PDL1 expression is predominantly quantified by immunohistochemistry (IHC) assays, and some recent research has also indicated a significant correlation between mRNA expression levels and the response of associated monotherapies[13]."

We would like to extend our thanks once again for your valuable comments and suggestions during the review process.

Referee 2

Thank you for your valuable comments and suggestions during the review process.

Referee 3

The authors have submitted a revised manuscript with multiple clarifications and including FFPE (in addition to fresh frozen) sample analysis. I find the authors responsive to reviewer feedback and the current manuscript far stronger (and more clear) than initial submission. My major concerns have been addressed.

Thank you for your valuable comments and suggestions during the review process.